# Individually unique, fixed stripe configurations of *Octopus chierchiae* allow for photoidentification in long-term studies

**Benjamin Liu**[☯], **Leo Song**[ORCID]*[☯], **Saumitra Kelkar, Anna Ramji, Roy Caldwell**

Department of Integrative Biology, University of California Berkeley, Berkeley, California, United States of America

☯ These authors contributed equally to this work.

* leosong@berkeley.edu

**Data Availability Statement:** All relevant data are within the paper and its Supporting information (Supplementary Materials) files.

## Abstract

The Lesser Pacific Striped Octopus, *Octopus chierchiae*, is a small iteroparous octopus known to inhabit intertidal regions of the Pacific coast of Central America. Many details about its life history and ecology remain unknown. For apparently rare and delicate animals such as *O. chierchiae*, non-extractive sampling methods are necessary to study individuals and populations over time. After photographically documenting the physical development of 25 octopuses from hatching, we have concluded that *O. chierchiae* has individually unique stripe configurations that remain constant throughout their post-hatchling lifetimes. Furthermore, using photographs taken of animals in captivity on different dates over many months, we show that untrained volunteers can accurately identify whether or not a pair of images depicts the same individual octopus. These results demonstrate that laboratory-reared individuals could be identified via photographs taken at different points in their lifetimes, which suggests wild individuals can also be recognized and observed for longitudinal field studies. In addition, our results imply potential for photoidentification and community science to be used as non-extractive, non-intrusive sampling methods for future studies of wild *O. chierchiae*.

## Introduction

Octopuses' remarkable camouflage ability, complex behavior, and intelligence have attracted vast scientific interest [1] and pop-culture intrigue [2]. However, their soft body forms, elusive behavior, and ever-threatened habitats make it difficult to track individuals in the wild over time [3]. Octopus body patterns are produced by a combination of pigment-filled chromatophores, color-reflective iridophores, and passively reflective leucophores, as well as muscular and hydrostatic forces that produce textural details [4, 5]. While these characteristics allow for rapid, neurally controlled color change, they operate within the fixed anatomical architecture of an octopus's skin [6, 7]. In some species, the consistency of body color patterns may allow humans to identify individuals from photographs, thereby aiding longitudinal studies [3]. *Octopus chierchiae* [8] exhibits qualities that may permit, and benefit greatly from,

**Funding:** The author(s) received no specific funding for this work.

**Competing interests:** The authors have declared that no competing interests exist.

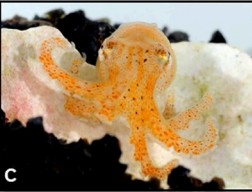

**Fig 1. Photographs depicting *Octopus chierchiae* of different ages.** (A) One-year-old *O. chierchiae* adult. (B) One-day-old *O. chierchiae* hatchling. (C) Five-day-old *O. chierchiae* hatchling already displaying a developing stripe configuration.

photoidentification. Also known as the Lesser Pacific Striped Octopus or the Pygmy Zebra Octopus, this species of dwarf octopus is found in shallow waters on the Pacific coast of the Americas, from Colombia to Baja California [9]. *Octopus chierchiae*–and the yet-undescribed Larger Pacific Striped Octopus and *Octopus zonatus* [10]–are commonly referred to as "harlequin" octopuses in reference to the black and white stripes and spots that cover their entire body [Fig 1A]. These stripes are not initially apparent in hatchlings [Fig 1B], but may be observed with magnification as early as the fifth day post-hatching [Fig 1C]. Although the animals may take on all pale or all dark morphologies, *O. chierchiae* most often display the high-contrast stripe-bar-spot morphology (defined in reference to the Larger Pacific Striped Octopus by Caldwell et al. [11] [Fig 2] where the stripe patterns are most clearly visible while the animal is at rest.

While most octopus species exhibit semelparity and only reproduce near the end of their lifespan [12], *O. chierchiae* is iteroparous [13]and have been observed to lay up to eight clutches of eggs over the course of their lifespan in our laboratory. Overall, *O. chierchiae's* iteroparous reproductive strategy [13], observed lifespan of over two years in our lab, and lack of a planktonic developmental stage [14] make breeding and maintaining the species in captivity a practical and viable pursuit, contributing to its growing potential as a model organism [15]. As its candidacy and status as a marine model organism develop, the need to locate, monitor, and study wild individuals of this species grows as well. Specifically, the ability to identify and observe the octopuses on an individual basis would be crucial for both field studies [16] as well as controlled laboratory research involving multiple individual animals [17].

Current studies on wild cephalopods most commonly involve tag-recapture studies [18], as exemplified by efforts to investigate the growth and movement of *Octopus vulgaris* in Central

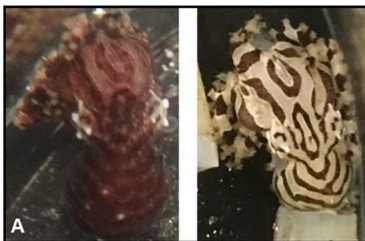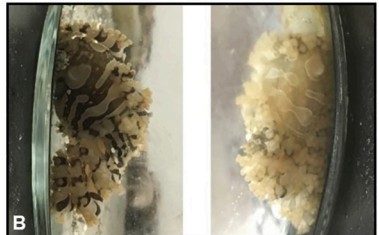

**Fig 2. Visibility of *O. chierchiae* stripe configuration during different body color morphs.** (A) *O. chierchiae* displaying dark morph (left) and stripe-bar-spot morph (right) illustrating how chromatophores activate directly over the leucophores to mask their whiteness. (B) A different *O. chierchiae* displaying stripe-bar-spot morph (left) and pale morph (right) illustrating how chromatophores expand adjacent to leucophore white markings to produce high-contrast markings.

Western Sardinia [19] and *Enteroctopus dofleini* in the Bering Sea [20]. Visible Implant Elastomer tags have demonstrated viable usage in long term studies of larger cephalopods such as *E. dofleini* [21]. However, as Semmens et al. [18] discuss, external tags can harm octopuses' delicate tissues, cannot be applied to animals that are too small, and can fall off or be deliberately removed by the tagged animal. Tattooing and branding techniques have had some success in cephalopod tracking studies as they eliminate the issue of tag removal and may be applied to small animals, but they are often practically difficult to execute and can cause the animal severe distress and damage [18]. Other rising technological tracking methods for cephalopods include chemical tagging, chemical analyses of cephalopod hard parts, geographic differences in parasite fauna, molecular genetics, and satellite data of detected signals [18]. However, *O. chierchiae*'s delicate and diminutive bodies (mantle length up to 4 cm) make longitudinal studies of wild individuals through the previously mentioned techniques difficult to complete or potentially dangerous to the animals.

Contrary to the aforementioned methods, photography serves as a largely inexpensive, non-invasive, non-extractive, and widely accessible technique to produce high-quality data. The low-impact nature of this approach as well as the virtual ubiquitousness of high-quality cameras that can record photographs, videos, geospatial data, and temporal data, has enabled community science efforts to make increasingly significant contributions to scientific research, as showcased by the US National Parks Service's What's Invasive! project and NSF NEON's Project BudBurst [22–24]. *In situ* research in particular has benefited tremendously from the growing availability of photodata, as many inferences on the abundance, distribution, phenology, and ecology of wild animal populations can be drawn from both the content and the context of these photographic images [22, 25], especially if members of the species of interest are individually and uniquely identifiable. The employment of such photography-analysis techniques has allowed researchers to estimate population dynamics [26], determine individual movement and home ranges [27] and quantify disease dynamics within populations [28]. Besides its exceptional accessibility, the photography-based approach offers yet another distinctive advantage: it may be applied retrospectively to existing photobanks [29]. All the aforementioned advantages along with the continual development of new computer-aided photoidentification methods [30–33] and analytical techniques make photography an ideal approach for monitoring and studying soft-bodied cephalopods [34–36] like *O. chierchiae*.

Longitudinal field studies conducted on the Caribbean reef squid, *Sepiotheuthis sepioidea*, have demonstrated that individual identification in the wild via body patterning and markings from injury can be viable in cephalopods [37]. However, as far as we know, *Wunderpus photogenicus* is the only octopus species in which individual cephalopods are shown to be identifiable by photographs of their individually unique stripe configurations [3]. Here, we test whether *O. chierchiae* also have identifiable, individually unique stripe configurations, and whether these remain constant throughout their lifetimes, allowing for photoidentification over long time spans by untrained volunteers.

## Materials and methods

### Ethics statement

This study was performed at the University of California, Berkeley in an AAALAC accredited facility, overseen by the Office of Laboratory Animal Care (OLAC), and thus follows the guidelines set forth by the Guide for the Care and Use of Laboratory Animals. Because the study species is an invertebrate, at the time of the study, neither USDA nor PHS policy covered this study and an IACUC approved animal use protocol was not needed. An official statement from the UC Berkeley OLAC is provided in the S1 File. Human participants were recruited via

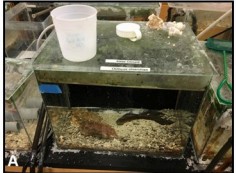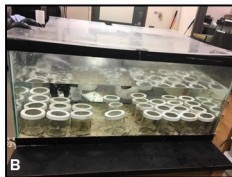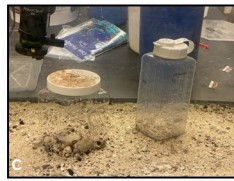

**Fig 3. Photographs depicting lab aquaria.** (A) One gallon (3.79 liter) individual mating tank where matings occurred and eggs were laid. (B) 40 gallon nursery tank where animals were reared in individual glass jars with screen lids from hatching until sexual maturity. (C) 20 gallon (75.71 liter) general tank where mature animals were housed in individual perforated plastic jars.

social media by reaching out in Slack channels and Facebook groups. These social media channels ranged from our workplace channels to hobbyist groups that are not biology or science oriented in order to minimize bias from participant backgrounds. All data were analyzed anonymously.

## Collection and rearing

Two male and two female adult *O. chierchiae* were obtained from a fully-licensed commercial collector in Nicaragua during 2017 and 2018. We housed them individually in one gallon (3.79 liter) glass aquaria lined with gravel, light sand, and bivalve shell fragments [Fig 3A] and provided PVC tubes, glass vials, or hollow rocks as shelter. Hatchlings were housed individually in glass jars with screen lids (9 cm tall, 8 cm diameter) containing a variety of substrates including dark sand, light sand, bivalve shell fragments, snail shells, and gravel, all within the same 40 gallon tank (151.42 liter) [Fig 3B]. Mature animals were housed individually in larger perforated plastic jars containing a similar mix of substrates, within a separate 20 gallon (75.71 liter) tank [Fig 3C]. As gonad maturity is difficult to assess in live cephalopods, we approximated the age indicative of sexual maturity to be 16 weeks old, based on behavioral observations, i.e. engaging in copulation. All aquaria in our laboratory were filled and maintained with artificial seawater (32–36 ppt) at 22–24˚C. Detailed breeding and rearing methods can be found in the (S2 File).

A total of 156 individuals from 10 clutches hatched over the course of this study (3 to 32 hatchlings per clutch) with an average body weight of 17.4 mg (min = 9.5 mg, max = 25.6 mg) (S1 Dataset). Comparisons of individuals from different clutches, hatching times, and parents were not tested in this study. Captive-bred *O. chierchiae* that survived the first few weeks of life typically lived for at least a year, with our longest-lived captive-bred octopus being a male hatched in July 2018 who lived for 22 months and fathered at least 4 different clutches of offspring before his death in June 2020. While the exact age and influence of life experiences on the development of wild caught individuals cannot be determined, having study animals bred and raised entirely in captivity allowed us the unique opportunity to follow the species' development throughout its entire lifecycle, from hatching until death.

## Photography

We photographed individual *O. chierchiae* from above water, using a handheld Apple iPhone 6s smartphone and a handheld 4K Sony camcorder (model HDR-XR260V) from June 2018 to July 2020. Due to high hatchling and juvenile mortality rate [15] and some individuals' aversion to open environments, only a subset [n = 25] of our lab reared animals were involved in this study. We only included individuals whose lifespan and activity patterns allowed stripe

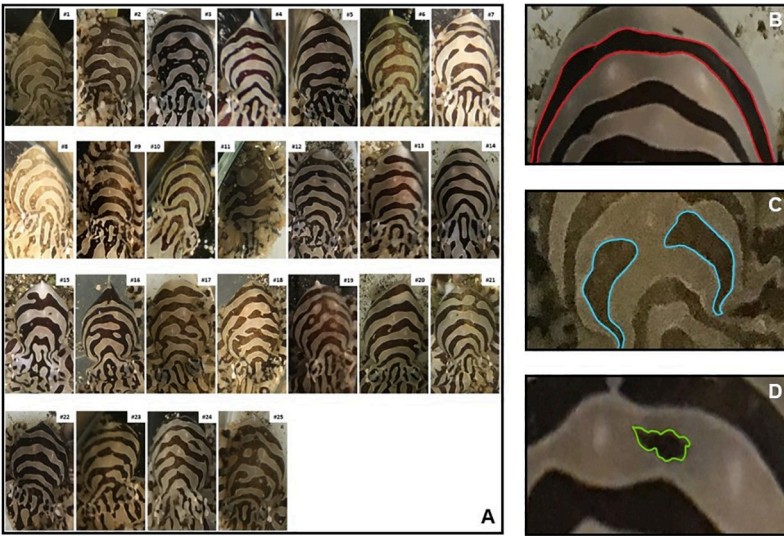

**Fig 4. Specific and polymorphic dorsal patterns on mantle of different individuals of *O. chierchiae*.** (A) Side-by-side display of dorsal mantle surface of all *O. chierchiae* individuals involved in the study. (B) Example of a continuous stripe [red]. (C) Example of a discontinuous stripe [blue]. (D) Example of a spot [green].

configurations to be clearly photographed on multiple occasions. In order to obtain a photographic record of each individual's development over the course of their lifetime, we photographed lab-reared *O. chierchiae* weekly, without removing them from their containers.

Terms used to describe *O. chierchiae* body patterns follow those defined and illustrated previously [6, 11, 38, 39]. Morphological appearance of each animal may vary greatly depending on body position or chromatophore activation. Consequently, this investigation focused on the stripe configurations of the chronic patterns located on the animals' dorsal mantle surface [Fig 4A], because these were the most consistently observable identifying characteristics. Chronic patterns are those that persist for hours or days, or those that the animal returns to after acute variations that last seconds or minutes [38]. The stripe-bar-spot pattern is generally composed of brown and white head bars, eye bars, extended eye bars, and transverse (chevron) stripes on the head and mantle, and longitudinal stripes on the arm crown. Occasionally, small brown mantle dark spots can be found within a white mantle stripe, and vice versa. The precise shape of each of these components was variable between individuals. We chose to specifically examine the dark transverse mantle stripes and dark mantle spots in the chronic patterns of *O. chierchiae* (S3 Dataset). We defined dark transverse mantle stripes as any dark pattern component that traverses the width of the individual's mantle. We further categorized these into dark continuous stripes [Fig 4B] and dark discontinuous stripes [Fig 4C], those that contained no visible breakage and those that contained visible breakage, respectively. We also defined dark mantle spots [Fig 4D] as any dark pattern component completely encircled by white pattern components on an individual's mantle. Although we provide above a list of body pattern component definitions, this study focused on the animal's body patterning as a whole. These components may come together in many combinations, including overlapping one another, to produce a wide variety of patterns unique to each individual animal [Fig 4A]. Each observer may focus on a different identifying characteristic (such as focusing on the pale components rather than the dark components) within the stripe configuration and yet still come to the same conclusion.

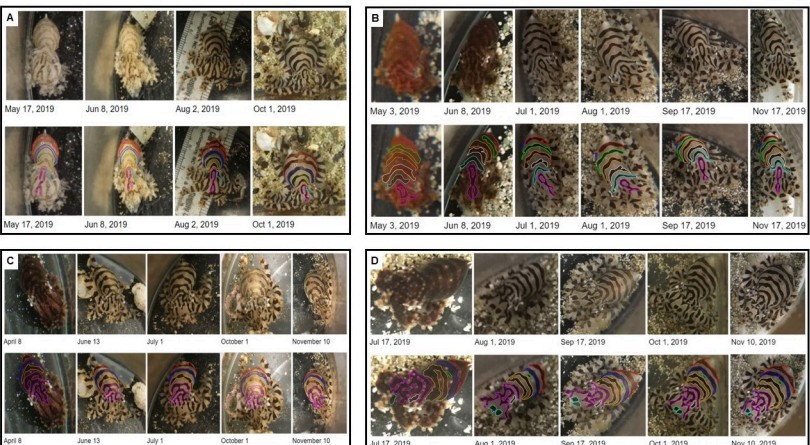

**Fig 5. Timeline of *O. chierchiae* individuals showing consistent body patternings.** The bottom row of each panel duplicates these photographs, with analogous pattern components color coded. (A) Timeline of *O. chierchiae* individual (hatched Nov 12, 2018) over a 5 month time span. (B) Timeline of *O. chierchiae* individual (hatched May 6, 2019) over a 6 month time span. (C) Timeline of *O. chierchiae* individual (hatched Nov 26, 2018) over a 7 month time span. (D) Timeline of Octopus individual (hatched May 6, 2019) over a 4 month time span.

Per our observations, disturbances could stimulate a change in the animal's color morph [Fig 2] or flashing between morphs. Over the course of our study, we attempted to avoid disturbing and distressing the animals during husbandry and photography sessions. If an individual was hiding or clinging to the jar lid, they were not photographed that session. One photograph per individual per session was taken from angles relative to the animal's body position at the time of viewing in order to capture the octopus' dorsal mantle surface as clearly as possible. For the four individuals with the most extensive photobanks, we arranged a series of photographs chronologically, one per month if available, and traced their stripe configurations using Adobe Illustrator. Each isolated pattern component was color coded [Fig 5] to visualize and establish the consistency of the stripe configurations. Body patterns of *O. chierchiae* became visible to the naked eye when individuals were around two weeks old and were consistently fully visible by four weeks, but can be observed with magnification as early as the fifth day after they hatch [Fig 1C].

## Survey design and evaluation

We created a survey following the methods of Huffard et. al. [3] using the photobank of images taken over the course of our study. The survey (S3 File) consisted of a series of 20 slides, arranged by a random number generator, each depicting two photographs of *O. chierchiae* individuals side by side, with no repeated measurements. Photograph pairs were rotated and cropped to display the animals in similar orientations and sizes. Only photographs of animals over four weeks old were used in the survey as that was the age when all the individuals' stripe configurations became clearly and consistently discernable by our photography equipment. We solicited survey participants via social media for two weeks, and collected responses via Google Forms for a total of four weeks. For each slide, participants were asked to respond whether the slide appeared as a "match" (two images of the same individual taken at different times) or "no match" (two images of different individuals). No time limit or built-in zoom function was implemented and participants were free to go back and edit responses prior to

submission. The survey contained nine slides depicting matching photographs, taken up to 25 weeks apart, and 11 slides depicting non-matching photographs. Each set of survey responses was given a score $\left[\frac{correct\ answers}{total\ questions} \times 100\right]$. Incorrect answers were categorized as false positive (meaning an erroneous match) and false negative (meaning a missed match). Unanswered questions were scored as incorrect, but omitted in our false positive-false negative analysis. In our social media solicitation, we reminded participants to complete the survey on a computer screen, as designed. However, because our data collection methods were not face-to-face and were analyzed anonymously, we cannot confirm the device each survey was completed on.

These survey designs were refined based on a preliminary survey detailed in (S4 File and S2 Dataset).

## Results

Over the course of the study, we observed that *O. chierchiae* individuals' [n = 25] stripe configurations were consistent throughout their lifetimes, evidenced by photographic records [Fig 5]. The *O. chierchiae* bred and raised in our lab displayed [Fig 4] an average of 4.8 (±0.49 S.D.) dark transverse mantle stripes in their chronic patterns. In 20% (n = 5) of the animals, this included one discontinuous stripe [Fig 4C]. Additionally, 24% (n = 6) of the animals also displayed a dark mantle spot [Fig 4D] (S3 Dataset). Examples of how pattern component counts were obtained may be found in S1 Fig. All members of our research group (the authors of this paper) were able to recognize individual animals based on the uniqueness [Fig 4] and consistency [Fig 5] of the octopuses' stripe configurations. Next, we investigated if untrained observers would be able to do the same. Our study focused on untrained volunteers specifically to control for potential overestimation from trained experts with daily exposure to the animals.

Survey respondents (n = 38 responses) had a mean score of 84.2% ± 15.4% standard deviation, with a standard error of the mean of 2.5%. Survey respondents had a median score of

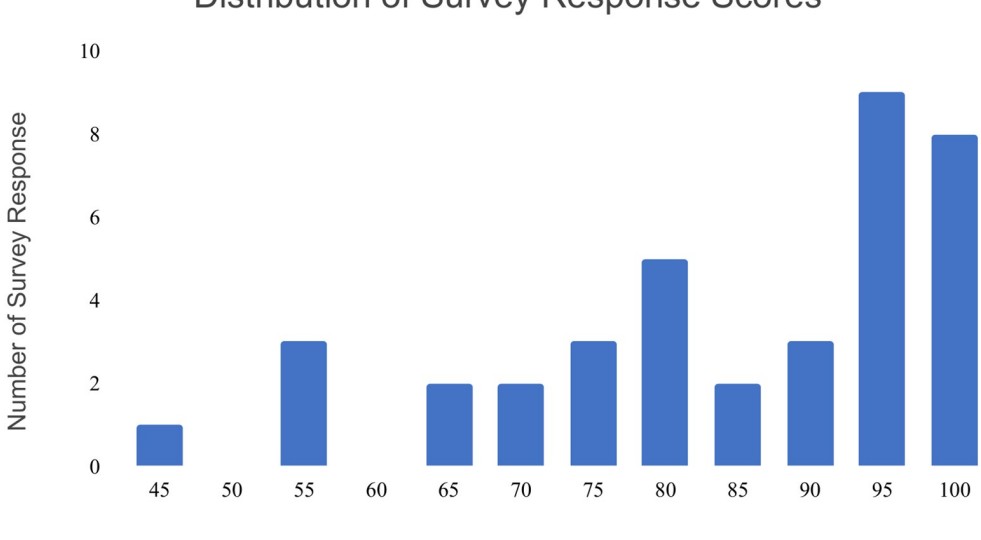

**Fig 6. Distribution of survey response scores (n = 38 survey responses).** Survey scores in percent are calculated by $\left[\frac{correct\ answers}{total\ questions} \times 100\right]$.

90%, with 44.7% of respondents (n = 17) scoring 95% or higher [Fig 6]. On average, respondents missed 13.7% of total possible matches (1.9 false negative responses out of 20 questions) and erroneously matched 17.2% of non-matching individuals (1.2 false positive responses out of 20 questions). Survey data can be found in the (S4 Dataset).

To investigate if there was a significant difference between false positive and false negative rates in the survey response, we performed the Wilcoxon Signed Rank Test with our null hypothesis being that false positive rates and false negative rates were the same, while our alternative hypothesis being that false positive rates and false negative rates were different. The test result revealed that there was not a significant difference between false positive and false negative rates (Wilcoxon Signed Rank Test: test statistic = 150, α = 0.05, n = 28, critical value = 116), suggesting a potential future study of wild populations would not be biased towards one error type or another.

## Discussion

We hypothesized that untrained survey participants would be able to identify *O. chierchiae* individuals based on their stripe configurations. This hypothesis was supported. The full group of participants had an average score of nearly 84.2%, while 52.6% of all participants scored at least 90% on the survey [Fig 7], showing that *O. chierchiae* individuals were distinguishable to a majority of untrained observers. The only participant to score 50% (expected value of random choice) or below responded to every question the same. Due to the anonymous nature of our methods, we are unable to confirm if this participant truly could not distinguish between the individuals in each photograph, or if they did not put forth genuine effort in responding to the survey questions. No individual question was answered incorrectly by a majority of participants. Based on these results, we conclude that *O. chierchiae* can be consistently identified throughout their lives via their individually unique stripe configurations by both trained and untrained observers, when an unobstructed, roughly in-focus view of the stripe configuration on the mantle's dorsal surface is provided.

Having hatched, reared and documented the individual *O. chierchiae* used in the surveys in captivity over extensive time spans–often over the course of their entire life history–we observed no evidence that stripe configurations changed over the lifetimes of

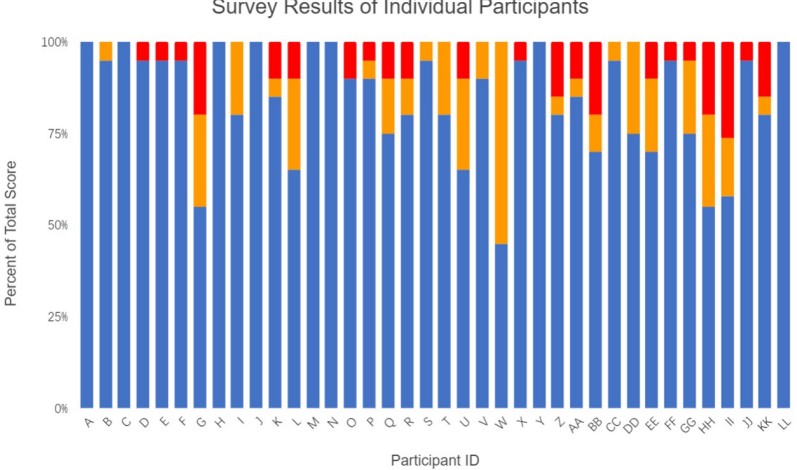

**Fig 7. Individual survey response scores (n = 38).** Bar chart illustrating total score (Blue), false negative (Red), and false positive (Orange) when assessing images.

individuals [Fig 5]. Our survey suggests that these permanent stripe configurations are a viable method of identifying the same individual at different points in time, even by volunteers that may have no experience with octopuses. To our knowledge, this is the first evidence that individually unique stripe configurations can be used by volunteers to consistently identify the same individual octopus throughout its development, from juvenile to adult.

This information could be used to design non-invasive studies monitoring populations of *O. chierchiae* over time *in situ*. It is conceivable that environmental disturbances and injuries may affect the stripe configuration of *O. chierchiae* in the wild. However, because so little is known about the species in its natural habitat and we worked to avoid injuring animals over the course of this study, we do not have conclusive evidence whether or not environmental conditions and injuries would affect an animal's stripe configuration.

Based on our results that participants could consistently identify whether two photographs depicted the same octopus, we recommend the survey design methods outlined above be employed in future community science efforts to study *O. chierchiae* via photoidentification. Furthermore, based on the high median scores recorded, it seems plausible that researchers may use this survey as a screener to identify a pool of volunteers capable of achieving highly accurate results. Smartphone apps or websites can be utilized to gather responses *en masse* and the process may even be automated via computer-assisted photoidentification software with enough responses and high quality image data. We envision untrained volunteers engaging in community science efforts by contributing to the establishment of a photodata repository. This could be done by sending a photograph in which the stripe configuration on the animal's dorsal mantle surface is visible along with the date, time, and geographic coordinates saved from a smartphone. Future research teams may make use of such a database and draw inferences about the species' natural history, including home range, behavior, population size, population density, and wild lifespan.

While it is known that there can be variation in the founder chromatophores of individual octopus hatchlings of the same species [40] and that *O. chierchiae* can begin to display the characteristic stripe pattern as soon as five days after hatching [Fig 1C], we did not begin photographing animals for this survey study until they were at least four weeks old. Thus, more work is needed to determine whether the individually unique stripe configurations of *O. chierchiae* are fixed from hatching, and if not, at what point in early development the stripe configurations become permanent. Identifying the specific anatomical composition of *O. chierchiae*'s stripe configuration was also beyond the scope of our study, and would have required sacrificing our study subjects. We encourage future research efforts to be directed into examining the skin histology and ultrastructure of *O. chierchiae*, when deceased individuals are available, in order to gain further insight into the physiological basis of their stripe configuration development.

Does *O. chierchiae*'s individually unique, lifelong body patterning play a role in intraspecific behavior in the wild? Octopus species are known to have the ability to recognize conspecifics [41]. Octopuses generally have high visual acuity, and the ability to remember [42]. Individual recognition has been explored in several other cephalopods [43, 44], and at least one species, *Octopus vulgaris*, is thought to have the ability of individual conspecific recognition [45]. The harlequin octopuses, with their bold semi-permanent body patterns, also "rely less upon changes in chromatophores than upon perceptions of postures and positions," [46]. In our view, these factors make it worthwhile to explore whether individual recognition based on visual cues plays a role in the lives of wild *O. chierchiae* and other octopus species.

## Supporting information

**S1 File. OLAC ethics statement.**
(PDF)

**S2 File. *O. Chierchiae*. breeding and rearing methods.**
(DOCX)

**S3 File. Survey.**
(PDF)

**S4 File. Preliminary survey and discussion.**
(PDF)

**S1 Dataset. *O. chierchiae* hatchling weights.**
(XLSX)

**S2 Dataset. Preliminary survey responses.**
(XLSX)

**S3 Dataset. Dark mantle pattern component counts.**
(XLSX)

**S4 Dataset. Survey responses.**
(XLSX)

**S1 Fig. Dark mantle pattern component counts example.**
(TIFF)

## Acknowledgments

This study would not have been possible without the hard work and support of the Caldwell Lab's *O. chierchiae* husbandry team over the past few years (2016–2020). Dr. Christine Huffard imparted much insight and expertise throughout this investigation and critiqued earlier drafts of the manuscript. Dr. Huffard's 2008 manuscript on *W. photogenicus* inspired this paper. Dr. Caroline Williams from UC Berkeley and Bret Grasse from the Woods Hole Marine Biological Laboratory provided correspondence and guidance. Richard Ross from the California Academy of Sciences contributed to husbandry methods.

## Author Contributions

**Conceptualization:** Benjamin Liu, Saumitra Kelkar, Roy Caldwell.

**Data curation:** Benjamin Liu, Leo Song.

**Formal analysis:** Benjamin Liu.

**Funding acquisition:** Roy Caldwell.

**Investigation:** Benjamin Liu, Leo Song, Saumitra Kelkar, Anna Ramji, Roy Caldwell.

**Methodology:** Benjamin Liu, Leo Song, Saumitra Kelkar, Anna Ramji, Roy Caldwell.

**Project administration:** Benjamin Liu, Leo Song.

**Resources:** Roy Caldwell.

**Software:** Benjamin Liu.

**Supervision:** Leo Song, Roy Caldwell.

**Validation:** Benjamin Liu, Leo Song, Saumitra Kelkar, Anna Ramji, Roy Caldwell.

**Visualization:** Benjamin Liu, Leo Song, Anna Ramji.

**Writing – original draft:** Benjamin Liu, Leo Song, Saumitra Kelkar.

**Writing – review & editing:** Benjamin Liu, Leo Song, Saumitra Kelkar, Anna Ramji, Roy Caldwell.

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
