## [Decision Letter · Decision Letter 0]

26 May 2022

PONE-D-22-05774Individually unique, fixed body patterns of *Octopus chierchiae* allow for photo-identification in long-term studiesPLOS ONE

Dear Dr. Song,

Thank you for submitting your manuscript to PLOS ONE. After careful consideration, we feel that it has merit but does not fully meet PLOS ONE’s publication criteria as it currently stands. Therefore, we invite you to submit a revised version of the manuscript that addresses the points raised during the review process. Both reviewers have made constructive comments that can help to improve the manuscript. Please submit your revised manuscript by Jul 10 2022 11:59PM. If you will need more time than this to complete your revisions, please reply to this message or contact the journal office at plosone@plos.org. Please include the following items when submitting your revised manuscript:A rebuttal letter that responds to each point raised by the academic editor and reviewer(s). You should upload this letter as a separate file labeled 'Response to Reviewers'.A marked-up copy of your manuscript that highlights changes made to the original version. You should upload this as a separate file labeled 'Revised Manuscript with Track Changes'.An unmarked version of your revised paper without tracked changes. You should upload this as a separate file labeled 'Manuscript'.

We look forward to receiving your revised manuscript.

Kind regards,

Erik V. Thuesen, Ph.D.

Academic Editor

PLOS ONE

Journal Requirements:

Reviewers' comments:

Reviewer's Responses to Questions

**Comments to the Author**

1. Is the manuscript technically sound, and do the data support the conclusions?

Reviewer #1: Partly

Reviewer #2: No

2. Has the statistical analysis been performed appropriately and rigorously? 

Reviewer #1: N/A

Reviewer #2: I Don't Know

3. Have the authors made all data underlying the findings in their manuscript fully available?

Reviewer #1: Yes

Reviewer #2: Yes

4. Is the manuscript presented in an intelligible fashion and written in standard English?

Reviewer #1: Yes

Reviewer #2: No

5. Review Comments to the Author

Reviewer #1: Liu and colleagues investigated if body patterns in Octopus chierchiae are individually unique and consistent over time and if untrained volunteers can distinguish individuals of this species from pictures. Wild collected octopus were reared and mated in laboratory. A subset of the F2 hatchlings was raised in laboratory and photographed weekly for two years. Untrained volunteers were subjected to an online survey showing pairs of pictures with and without superimposed pattern tracers to test if they were able to distinguish among different individuals. The authors concluded that O. chierchiae showed individually distinctive stripes patterns that are stable through most of its lifespan, opening to the possibility of using them for field photoidentification without animal manipulation, and that non-experts can distinguish among specimens successfully when a clear view of body patterns is provided.

This study addresses the crucial issue of enabling studies on animals without compromising their welfare and I highly appreciate the effort of establishing a minimally/not invasive identification method. Such an approach can pave the way to numerous, exciting applications in this and other species that are currently largely overlooked due to the technical challenges associated with handling fragile organisms. Despite my interest and positive attitude towards such a kind of studies, I would encourage the authors to be clearer and more detailed in their methodological descriptions.

In the current form, I am unable to evaluate if the chosen methodology is appropriate and I have concerns on data analysis and interpretation. If the points indicated below are addressed, then it would be easier to understand their experimental and analytical choices, the presented results, their interpretation and hopefully they will help in making the conclusions and manuscript more solid.

Specific comments

Aims and Methods: to my understanding and as anticipated above, this study: tests 1) if stripes patterns can be used as individual identifiers across lifetime in this species; and 2) if untrained volunteers can distinguish between individuals through photos.

First, it seems to me that the uniqueness and consistency of body patterns over time was not tested in an explicit framework. The main text did not mention any statistical analyses to test these hypotheses (uniqueness and consistence, which are two separate features that should not be confused), only a reference to personal observations and pictures (lines 201-204). These results need to be supported by a more solid and objective evidence on the distinctiveness and stability of patterns. For example, individual and time-series pictures could be compared through computer-assisted photo identification software or a survey like the one for untrained volunteers performed on experts. Why were identification software not considered in this study nor in future perspectives? They could be helpful in discerning patterns.

Second, it is not clear to me why the authors are interested on photoidentification from untrained volunteers specifically. Was it a control to avoid potential overestimation from trained experts? If this was the case, it should be written explicitly. According to the main text (lines 108-109, 178-189, 204-213, 216-218, 246-256) it seems that the authors are mainly considering some kind of a citizen science project. Such projects largely differ in type of collected data, required expertise, bias, etc. For example, a project where an untrained volunteer sees an octopus while swimming, she/he takes a photo, sends it together with geographic coordinates taken from smartphone to an expert (i.e., the volunteer provides raw data that is then analyzed by an expert) is different from a project where a person is simply asked if she/he saw the same specimen (i.e., the volunteer “analyses” data on her/his own). The test to explore if a citizen science project can be effective should consider and reflect these differences, starting from how pictures are taken and who analyze them (volunteer? Expert?). More detailed information on the envisaged contribution of untrained volunteers should be added and the analyses valuated accordingly, I am largely unable to assess them in the current form.

Lines 154-162: more details on how samples were photographed are needed to understand the study design. For example, why were not all the hatched individuals photographed? Why was a sample size of 25 chosen for this analysis? How were the photographed samples selected?

Lines 144-146: mentioned multiple clutches and Figure 4 indicates different hatching time (November 2018, May 2019), were any differences in body pattern uniqueness and consistency tested among clutches, hatching time and parents?

Were the cameras underwater or emerged? Both settings should be tested if both possibilities could occur in the citizen science project, and the limits associated with such techniques should be discussed (e.g., a picture taken by an emerged camera might be distorted due to refraction and waves).

Were pictures standardized (e.g., taken from the same height through a copy stand) and repeated at each session and view?

Lines 158-160 not fully clear: how many pictures were taken for each individual and session? One top view plus two side view pictures?

Lines 179-185: how many slides were shown to participants? Were slides designed to be viewed in a computer monitor or smartphone? Was a zoom function, possibility to go back and edit a response or time limit included? Were entries randomized (e.g., to avoid showing all matches first)? Were repeated measurements (e.g., presenting the same photos pair multiple times) included?

Lines 181-182: why the maximum age gap between the two pictures in the same slide was five months? Did the zero days apart photos represent the repeated pictures of the same individual on the same day with the same settings or different ones (e.g., pictures taken at different time, or a side and a top view)?

Lines 185-189: while more attention was dedicated to the survey description, what and how the collected responses were analysed is less clear. For example, the meaning of “compiled” and “graded”, and most importantly, the hypotheses that were tested using the t-tests (please explicitly mention it and its methodological reference), the compared groups and the analyzed data/variables are not obvious. Was data analysed anonymously? The t-test is a parametric approach with specific assumptions to meet but it looks like that these were not tested. If dichotomous data was analyzed, the t-test might not be the most appropriate approach.

Lines 205-239: the number of surveyed untrained volunteers (22) seems to be quite low. How long was the survey open? How was the sample size chosen?

Such surveys can be biased by demographic factors such as participants’ age, sex, education, occupation, geographic provenance, etc. Was this type of information collected for each participant? Responses should be evaluated in the light of these factors. Issues in vision and the use of a computer vs. smartphone screen might explain some inter-individual variance in responses too. Additionally, the fact that some participants responded to both surveys (line 239) introduce a potential bias that need to be considered.

Lines 216-226: in addition to the concerns highlighted above, I do not think that the conclusion that this species can be consistently identified by untrained observers can be supported so strongly without at least testing if the percentage of correctly assigned matches increases when the issues identified in lines 221-223 are fixed. An explicit comparison between non-expert and expert performance and a measurement of identification accuracy would be more informative too. These analyses would make this study substantially more solid and comprehensive.

Line 242-246: this study potentially (please see my comments about testing uniqueness and consistency of body patterns above) showed that stripe patterns are stable over time in captivity in a fully standardized rearing environment since before birth. Could variable field conditions and injuries affect body patterns? This and similar potential limits of these results should be discussed.

Lines 220-226: the “specific examination of the problematic questions” should be described in more details. How were the factors underlying the problematic questions identified? Was their influence explicitly tested or just hypothesised?

Ethics statement: several key information such as the field permit number (or if this is not needed or it is included in the formal waiver already), formal waiver number and details on animal welfare are missing. Research projects involving human participants includes the use of data collection methods which are not face-to-face such as online surveys unless they are analysed anonymously, but it is not clear if this was case. The appropriate details should be added here and in Materials and Methods as indicated by the journal.

Minor comments

Line 31: here and in the rest of the text, I would be more specific in indicating that body patterns refer to stripes’ configurations, as this study did not consider other features such as skin texture.

Line 34: the untrained volunteers survey was conducted on laboratory-reared animals, not wild ones. Please rephrase this sentence and the rest of the main text accordingly. Please also consider replacing “demonstrate” with “might suggest”.

Line 87-88: I would suggest replacing “innocuous” with “non-invasive”. I agree that data collection through photoidentification is less invasive than handling, but it likely still involves some stress, being observed/photographed by someone can be an unpleasant experience for humans too. I would avoid using strong words such as “harmless”, “non-intrusive” and instead indicate that such techniques are less impactful on animal welfare.

Line 35, 92, 251, 263: “in situ“ should be in italics. Alternating or replacing it with “field” might be clearer to readers.

Line 102: please add more recent references for photo identification methods.

Line 134: what is the age of sexual maturity?

Lines 140-141: please explain the relevance of analysing a generation bred and raised in captivity.

Lines 141-146: do these numbers refer to the F2 alone or include the F1?

Lines 136-137: are all mates from the F1 generation?

Lines 179-180: please specify if “this photobank” refers to pictures collected in this study or in the mentioned reference.

Line 185: the pattern tracers were superimposed, not below the photo. Please be aware that some visually impaired (e.g., colour-blind) people might have difficulties in seeing them.

Lines 191-198: the main text, specifically the Methods, should include this important information that is currently disclosed only these captions, i. e., the pictures pairs showed either the same individual on different days or different individuals on the same day.

Lines 202-204: please specify if experts were able to recognize individuals thanks to the uniqueness or/and consistency of body patterns.

Line 240: the survey C was not mentioned before.

267-268: please report all the information in Methods, including the age at which photo recording started. Why were animals not photographed until they were at least four weeks old?

Discussion: I would be more cautious in using expressions such as “our surveys confirm” (line 244), “suggest” would be more appropriate.

Line 255: “en masse” should be in italics.

Lines 250-287: I enjoyed reading these potential applications, which included a width of perspectives that should be more common in methodological studies.

Figure 1 and 2 caption: the species’ name should not start with a capital letter and the complete name should be in italics, as reported elsewhere in the manuscript.

Figure 1-2 and 6-9: I would suggest grouping these pictures in fewer figures (e.g., Figure 1 and 2 could become a single one with multiple panels).

Figure 10 and 11: these figures and captions mentioned “scores” that were not explained in the main text. Please describe how these “scores” were obtained from the surveys’ responses in Methods.

The number of figures is quite high and might overwhelm readers. I would suggest focusing on few, most relevant ones and moving the others to Supplementary.

Data availability: please include a statement in the main text too and specify what type of data is available.

Numbers from zero to ten (in some cases to thirteen) are conventionally written using words and not numbers.

Reviewer #2: The idea behind this paper is worthwhile yet the scholarship overall is subpar and there is one major problem. Both can be remedied if the authors concur - and want their research to be used and further developed by others. I provide examples below and give suggestions on how to address these. For example, the authors provide a very slanted choice of references to “support” their paper (sample details below).

The major flaw in the paper is that the authors do not understand that leucophores in the dermis are the basis of these permanent and species-specific white markings; in this case of O. chierchiae they are also the basis of individual specificity. Leucophores are not even mentioned in this paper.

Here is the basic scenario: chromatophores are mostly evenly distributed in the skin and when selectively activated create pigmentary patterns. Iridophores and leucophores (i.e. “white cells”) are not evenly distributed; they occur only in specific parts of the skin. Leucophores are aggregates of static cells (they do not have nerves, muscles etc) and are common in nearly all octopuses and cuttlefish. Andrew Packard and his colleagues as well as Roger Hanlon and colleagues have published widely on this and you need to reference some of those papers so that readers understand where this “fingerprinting” comes from. (For example: Packard, A., & Hochberg, F. G. (1977). Skin patterning in Octopus and other genera. Symposia of the Zoological Society of London, 38, 191-231. )

Both of those authors have shown that innervation patterns of chromatophores are aligned to expand adjacent to leucophore white markings to produce the high-contrast zebra-like markings. This was demonstrated neurophysiologcally and anatomically, for example, in Sepia officinalis (see Figs in Hanlon and Messenger 1988, Phil Trans). Conversely, other neural command neurons activate chromatophores directly over the leucophores to diminish (or mask) their bright whiteness.

By the way, leucophores produce some of the whitest white known in the animal kingdom (Mäthger, L. M., Senft, S. L., Gao, M., Karaveli, S., Bell, G. R. R., Zia, R., . . . Hanlon, R. T. (2013). Bright white scattering from protein spheres in color changing, flexible cuttlefish skin. Advanced Functional Materials, 23(32), 3980-3989 ) so this precise pigmentary masking by chromatophores is important and O chierchiae do this as demonstrated by your images.

Your Supplementary Fig. 10 in file S2_file.pdf shows this very well !! This figure has to be elevated to the main manuscript to illustrate what I explained above.

Some specific comments:

INTRODUCTION

Line 41 - delete this sentence - it adds nothing to the paper and such a statement applies to thousands of papers. Furthermore, the reference for this is inappropriate as it is written by someone who only studies octopuses.

L 47 the Hackett reference only covers the pop culture side of what is stated in this sentence. For the biological attributes supply a global rigorous study review such as Hanlon and Messenger 2018 book Cephalopod Behaviour or similar.

L49. There are numerous studies have tracked individual cephalopods in the wild over time. It does not matter that it is difficult. You should cite 2-4 such studies - that have been done on octopuses, cuttlefishes, squids and nautilus!

(One of many examples: Hanlon, R. T., Forsythe, J. W., & Joneschild, D. E. (1999). Crypsis, conspicuousness, mimicry and polyphenism as antipredator defences of foraging octopuses on Indo-Pacific coral reefs, with a method of quantifying crypsis from video tapes. Biological Journal of the Linnean Society, 66(1), 1-22. Retrieved from <go isi="" to="">://000078601200001

Overall this intro paragraph is heavily slanted and even a bit misleading. Kindly pose the study in a more appropriate and professional manner.

L 57. An appropriate reference for cephalopod semelparity is Rocha, F., Guerra, A., & Gonzalez, A. F. (2001). A review of reproductive strategies in cephalopods. Biological Reviews, 76(3), 291-304. doi:10.1017/s1464793101005681

L59 This statement of “… apart from most other octopus species” is nonsense. A very large number of octopus species produce very large eggs that hatch out as miniature adults (i.e. no paralarval/planktonic stage). The authors seem unaware of cephalopod life histories.

L66 The Grearson et al 2021 is not even published … it is BioRivX … Moreover, this species is anything but a model organism! There are only a handful of journal papers and the species is nearly impossible to obtain from the wild, and only recently have lab trials began in earnest. This sort of hyperbole is unacceptable.

L 102 There are literally dozens of papers in which cephalopods have been studied with photography; at least cite a few of them to justify this paper and this Introduction.

L 104 Another method to follow individual octopus, squid and cuttlefish is scar tissue in the skin, or missing arm(s), or even marking the skin with dye in specific spots. These need to be mentioned for proper context in this paper.

METHODS

This section if very long and includes many items that have nothing to do with this study. If you want credit for reading this species then do a separate ms and put in an appropriate mariculture journal. Otherwise shorten this section substantially. There are way too many figures in the main ms; if you choose to include so much culture info (which is not the biological basis of this paper) then at least put them in Supplementary sections.

RESULTS

L 201. How many individual octopuses were studied? The only hint of this is in the captions of Figs 7-9; from these the estimate would be perhaps as few as 2 octopuses. This is of central importance and the authors have to be crystal clear on this.

L 203 Where are the data on the results of the research group (this number has to be clear - do you mean the 5 coauthors?)

Fig. 10 text/number letters are way too small. The y axis title of Percent of Total Score is ambiguous - can you provide a more intuitive title so that the graphic can stand alone in terms of understanding? … to that point, a color legend could/should be put directly on the graph … there is ample space amidst the blue bars.

An important omission: you need to verify that the white stripes are leucophores and that their distribution is distinctive. The simplist fast method is to do some Light microscopy of gross anatomy of the leucophore stripes. This will give authenticity to your study. You can also illustrate this by taking a skin sample and photograph it normally then with light behind it to show that the leucophores stand out and block light in specific patterns. See Figs 14,15 i (Hanlon, R. T., & Messenger, J. B. (1988). Adaptive coloration in young cuttlefish (Sepia officinalis L.): The morphology and development of body patterns and their relation to behaviour. Philosophical Transactions of the Royal Society of London B, 320, 437-487. Note also Figs 18,19 the same paper to show how chromatophore expansion either accentuates or masks the whiteness.

DISCUSSION

L 217. These results are not very convincing - 75% correct is only fair to begin with, but even then only a bit more than half even scored that high.

Moreover, the whole treatise seems to be based on 2 (or a very few) octopus individuals. (See comments in Results; perhaps there is a better explanation about how many octopuses were analyzed in this study).

L 225 “when a clear view of the body patterns is provided.” This is a large caveat to the study - your images in the lab were not all that useful/clear either, so going on later to say this could be useful in the field is a quantum leap in difficulty and utility.

Overall, it appears that using such body patterns is quite difficult and requires considerable study and diligence by the biologists. The authors never really say this in this paper. Need to be more straight up on what the results actually indicate.

L 250+ Applying this technique to following individuals underwater will be very difficult (this reviewer has studied many octopus species in the wild; with many volunteers). The photography techniques in situ will require great diligence and patience to even get sufficient images to verify identity, much less acquire quantitative behavioral data.

L 244 To say you have no evidence of pattern change over time is a strong statement given how few animals were used.

L 245-249 Workers with some octopus and cuttlefish species have often used White frontal spots in octopus and details of zebra stripes in Sepia officinalis as species markers, and I believe J Boal made a comment on individual identity in S. officinalis in one of her pubs (you can search that one). Better off to give broad story and include leucophore markings in other cephs not just your octopus species.

This paper has potential but requires major revision. These comments are given in a constructive manner to improve the science.</go>

6. PLOS authors have the option to publish the peer review history of their article (what does this mean?). If published, this will include your full peer review and any attached files.

Reviewer #1: No

Reviewer #2: No

---

## [Author Response · Author response to Decision Letter 0]

9 Oct 2022

Editor comments:

Reply: 

 To our knowledge we have followed PLOS ONE's style requirements exactly.

Reply:

 We would like to clarify that we received no specific funding for this work and thus do not have any grant numbers or awards to report.

Reviewer #1: 

1. Liu and colleagues investigated if body patterns in Octopus chierchiae are individually unique and consistent over time and if untrained volunteers can distinguish individuals of this species from pictures. Wild collected octopus were reared and mated in laboratory. A subset of the F2 hatchlings was raised in laboratory and photographed weekly for two years. Untrained volunteers were subjected to an online survey showing pairs of pictures with and without superimposed pattern tracers to test if they were able to distinguish among different individuals. The authors concluded that O. chierchiae showed individually distinctive stripes patterns that are stable through most of its lifespan, opening to the possibility of using them for field photoidentification without animal manipulation, and that non-experts can distinguish among specimens successfully when a clear view of body patterns is provided.

This study addresses the crucial issue of enabling studies on animals without compromising their welfare and I highly appreciate the effort of establishing a minimally/not invasive identification method. Such an approach can pave the way to numerous, exciting applications in this and other species that are currently largely overlooked due to the technical challenges associated with handling fragile organisms. Despite my interest and positive attitude towards such a kind of studies, I would encourage the authors to be clearer and more detailed in their methodological descriptions.

In the current form, I am unable to evaluate if the chosen methodology is appropriate and I have concerns on data analysis and interpretation. If the points indicated below are addressed, then it would be easier to understand their experimental and analytical choices, the presented results, their interpretation and hopefully they will help in making the conclusions and manuscript more solid.

Reply:

 We appreciate the reviewer’s comments on our manuscript. We have incorporated further details about our methods and analysis in accordance with the reviewer’s suggestion and hope that our revision has made our study more interpretable.

2. “Aims and Methods: to my understanding and as anticipated above, this study: tests 1) if stripes patterns can be used as individual identifiers across lifetime in this species; and 2) if untrained volunteers can distinguish between individuals through photos.

First, it seems to me that the uniqueness and consistency of body patterns over time was not tested in an explicit framework. The main text did not mention any statistical analyses to test these hypotheses (uniqueness and consistence, which are two separate features that should not be confused), only a reference to personal observations and pictures (lines 201-204). These results need to be supported by a more solid and objective evidence on the distinctiveness and stability of patterns. For example, individual and time-series pictures could be compared through computer-assisted photo identification software or a survey like the one for untrained volunteers performed on experts. Why were identification software not considered in this study nor in future perspectives? They could be helpful in discerning patterns.”

Reply:

 We appreciate the comments and concerns raised by the reviewer.

Because access to this species is extremely limited, there is not a significant population of experts outside of our own research team available to perform the survey on and compare with the results from untrained volunteers. 

Although interesting, computer vision and other such computer-assisted photoidentification analytics were beyond the scope of expertise and resources of this research team at the time of the study. We proposed such studies making use of this technology be conducted on O. chierchiae in the future: 

Lines 302-304: “Smartphone apps or websites can be utilized to gather responses en masse and the process may even be automated via computer-assisted photoidentification software with enough responses and high quality image data.”

Furthermore, there is not an existing extensive repository of photographs for this species, making training computer-assisted photo identification tools unfeasible at the time. Often machine learning libraries rely on thousands to hundreds of thousands of images, and even then are not necessarily as effective as humans in finding matches.

However, inspired by the methods of Huffard et al. (2008) and Byrne et al. (2010), we performed traces of O. chierchiae stripe configurations over extensive time spans [Fig 4] and found that the pattern traces remained the same, as was the case for a single Wunderpus photogenicus tracked over time (Huffard et al. 2008). For this reason, we believe the conclusion that O. chierchiae have unique and consistent stripe configurations over time we reached with our study is valid and legitimate.

Lines 195-199: “For the four individuals with the most extensive photobanks, we arranged a series of photos chronologically, one per month if available, and traced their stripe configurations using Adobe Illustrator. Each isolated stripe was color coded [Fig 4] to visualize and establish the consistency of the stripe configurations. ”

3. Second, it is not clear to me why the authors are interested on photoidentification from untrained volunteers specifically. Was it a control to avoid potential overestimation from trained experts? If this was the case, it should be written explicitly. 

Reply:

We have since added this statement explicitly:

Lines 245-246: “Our study focused on untrained volunteers specifically to control for potential overestimation from trained experts with daily exposure to the animals.”

4. According to the main text (lines 108-109, 178-189, 204-213, 216-218, 246-256) it seems that the authors are mainly considering some kind of a citizen science project. Such projects largely differ in type of collected data, required expertise, bias, etc. For example, a project where an untrained volunteer sees an octopus while swimming, she/he takes a photo, sends it together with geographic coordinates taken from smartphone to an expert (i.e., the volunteer provides raw data that is then analyzed by an expert) is different from a project where a person is simply asked if she/he saw the same specimen (i.e., the volunteer “analyses” data on her/his own). The test to explore if a citizen science project can be effective should consider and reflect these differences, starting from how pictures are taken and who analyze them (volunteer? Expert?). More detailed information on the envisaged contribution of untrained volunteers should be added and the analyses valuated accordingly, I am largely unable to assess them in the current form.

Reply:

We have provided a detailed description of a potential way untrained volunteers would be able to contribute to community science efforts in monitoring and studying the species:

Lines 304-310: “We envision untrained volunteers engaging in community science efforts by contributing to the establishment of a photodata repository. This could be done by sending a photograph in which the stripe configuration of the animal’s mantle’s dorsal surface is visible along with the date, time, and geographic coordinates saved from a smartphone. Future research teams may make use of such a database and draw inferences about the species’ natural history, including home range, behavior, population size, population density, and wild lifespan.”

5. Lines 154-162: more details on how samples were photographed are needed to understand the study design. For example, why were not all the hatched individuals photographed? Why was a sample size of 25 chosen for this analysis? How were the photographed samples selected?

Reply:

We have included a more detailed description of our study design under the Photography Methods section of our manuscript:

Lines 180-184: “Due to high hatchling and juvenile mortality rate (Grearson et al., 2021) and some individuals’ aversion to open environments, only a subset [n = 25] of our lab reared populations were involved in this study. Only individuals with sufficiently long lives whose activity patterns allowed frequent visibility of stripe configurations to be clearly photographed on multiple occasions were included in this subset.”

 Lines 218-222: “We created a survey following the methods of Huffard et. al. (2008) using the photobank of images taken over the course of our study. The survey [S4 File] consisted of a series of 20 slides, arranged by a random number generator, each depicting two photographs of O. chierchiae individuals side by side, with no repeated measurements. Photo pairs were rotated and cropped to display the animals in similar orientations and sizes.”

6. Lines 144-146: mentioned multiple clutches and Figure 4 indicates different hatching time (November 2018, May 2019), were any differences in body pattern uniqueness and consistency tested among clutches, hatching time and parents?

Reply:

We have since included the following statement:

Lines 163-164: “Comparisons of individuals from different clutches, hatching times, and parents were not tested in this study.” 

7. Were the cameras underwater or emerged? Both settings should be tested if both possibilities could occur in the citizen science project, and the limits associated with such techniques should be discussed (e.g., a picture taken by an emerged camera might be distorted due to refraction and waves).

Reply:

We have clarified in Line 178: “We photographed individual O. chierchiae from above water…”

We appreciate the concern raised by the reviewer. However, expert advisors with field experience (Dr. Christine Huffard) have suggested that no practical difference would arise from emerged or submerged cameras. Because these octopuses are so small, the camera would need to be very close in order to take a picture, and waves/refraction would not be an issue. Additionally, since we are primarily concerned with the stripe configuration of the animals specifically, distortion due to refraction and waves would be of minimal concern. Even if the length and widths of each stripe is changed, the stripe configuration will be the same. Finally, waves and refraction were not issues in the study of photoidentification in W. photogenicus (Huffard et al. 2008), which relied heavily on photographs taken underwater by professional and recreational divers. 

8. Were pictures standardized (e.g., taken from the same height through a copy stand) and repeated at each session and view?

Lines 158-160 not fully clear: how many pictures were taken for each individual and session? One top view plus two side view pictures?

Reply:

We have clarified in Lines 191-195: “Over the course of our study, we attempted to avoid disturbing and distressing the animals during feeding, tank transfer, photography sessions, etc.. If an individual was sheltering or clinging to the jar lid, they were not photographed that session. One picture per individual per session was taken from angles relative to the animal’s body position at the time of viewing in order to capture the octopus’ dorsal mantle surface as clearly as possible. “

9. Lines 179-185: how many slides were shown to participants? Were slides designed to be viewed in a computer monitor or smartphone? Was a zoom function, possibility to go back and edit a response or time limit included? Were entries randomized (e.g., to avoid showing all matches first)? Were repeated measurements (e.g., presenting the same photos pair multiple times) included?

Reply:

 We have since provided more details about the survey design of our revised survey.

Lines 219-222: “The survey [S4 File] consisted of a series of 20 slides, arranged by a random number generator, each depicting two photographs of O. chierchiae individuals side by side, with no repeated measurements. Photo pairs were rotated and cropped to display the animals in similar orientations and sizes. ”

Lines 228-229: “No time limit or built-in zoom function was implemented and participants were free to go back and edit responses prior to submission. .”

 Lines 234-237: “In our social media solicitation, we reminded participants to complete the survey on a computer screen, as designed. However, because our data collection methods were not face-to-face and were analyzed anonymously, we cannot confirm if all the survey responses were completed on a computer screen or on any other device”

10. Lines 181-182: why the maximum age gap between the two pictures in the same slide was five months? Did the zero days apart photos represent the repeated pictures of the same individual on the same day with the same settings or different ones (e.g., pictures taken at different time, or a side and a top view)?

Reply:

Photo pairings for the survey were chosen by a random number generator. As a result, the maximum five month age gap between two pictures in our original survey was completely arbitrary and random.

Zero days apart was intended to represent photographs of two different individuals taken on the same day. To avoid confusion, the “zero days apart” description has since been deleted.

We have clarified our photography methods in Line 193-194: “One picture per individual per session was taken…”

11. Lines 185-189: while more attention was dedicated to the survey description, what and how the collected responses were analysed is less clear. For example, the meaning of “compiled” and “graded”, and most importantly, the hypotheses that were tested using the t-tests (please explicitly mention it and its methodological reference), the compared groups and the analyzed data/variables are not obvious. Was data analysed anonymously? The t-test is a parametric approach with specific assumptions to meet but it looks like that these were not tested. If dichotomous data was analyzed, the t-test might not be the most appropriate approach.

Reply:

We have clarified the descriptions of our collected responses: 

 Lines 231-234: “Each set of survey responses was given a score [(correct answers)/(total questions) × 100]. Incorrect answers were categorized as false positive (meaning an erroneous match) and false negative (meaning a missed match). Unanswered questions were scored as incorrect, but omitted in our false positive-false negative analysis.”

 Survey results data were analyzed anonymously, as described in Lines 235-237: “However, because our data collection methods were not face-to-face and were analyzed anonymously, we cannot confirm if all the survey responses were completed on a computer screen or on any other device.”

We have removed the t-test from our study as, per reviewer comment, it may not be appropriate for our data and adds little to the overall interpretation of our results. In its place we simply looked at the mean and median survey scores to draw our conclusions. 

Additionally, we have since clarified the hypothesis in Lines 257-259: “We hypothesized that participants would be able to identify O. chierchiae individuals based on their stripe configurations, meaning they would score notably above 50%, the expected value if responses were chosen randomly.” 

12. Lines 205-239: the number of surveyed untrained volunteers (22) seems to be quite low. How long was the survey open? How was the sample size chosen?

Reply:

 In our newly revised survey, 38 untrained volunteers participated in the survey. 

 The survey was promoted through social media solicitation for two weeks and was kept open for another two weeks for a total of a month as described in Lines 224-226: “We solicited participants via social media for two weeks, and collected responses via Google Forms for a total of four weeks.”

 The survey response sample size was not not chosen intentionally. We simply included all the survey responses that we received over the span of a month (the timespan which the survey was open for) and incorporated them into our study. We believe that a sample size of 38 is adequate and sufficient as similar survey based studies have included far fewer participants (Huffard et al. 2008; Byrne et al. 2010).

13. Such surveys can be biased by demographic factors such as participants’ age, sex, education, occupation, geographic provenance, etc. Was this type of information collected for each participant? Responses should be evaluated in the light of these factors. Issues in vision and the use of a computer vs. smartphone screen might explain some inter-individual variance in responses too. Additionally, the fact that some participants responded to both surveys (line 239) introduce a potential bias that need to be considered.

Reply:

 The survey was completely anonymous and no personal information of the participants was collected. 

 We agree with the concern raised by the reviewer regarding the use of differently sized electronic screens impacting participant’s survey response and have discussed it at length in Lines 234-237: “In our social media solicitation, we reminded participants to complete the survey on a computer screen, as designed. However, because our data collection methods were not face-to-face and were analyzed anonymously, we cannot confirm if all the survey responses were completed on a computer screen or on any other device.”

 In our new round of surveys, only one survey format is available, thus eliminating the potential bias introduced from participants responding to both of our original surveys.

14. Lines 216-226: in addition to the concerns highlighted above, I do not think that the conclusion that this species can be consistently identified by untrained observers can be supported so strongly without at least testing if the percentage of correctly assigned matches increases when the issues identified in lines 221-223 are fixed. An explicit comparison between non-expert and expert performance and a measurement of identification accuracy would be more informative too. These analyses would make this study substantially more solid and comprehensive.

Reply:

In our newly designed survey, all of the photographs selected were in-focus pictures of octopuses in their light morph with their stripe configurations on their mantle fully exposed and visible, thus eliminating the issues present in our previous survey.

 Like the reviewer, we were also profoundly intrigued by the idea of an explicit comparison between non-expert and expert performance and a measurement of identification accuracy. However, due to the incredible difficulty accessing the species, few individuals in the world have had extensive and sufficient experience working with the animal to be considered experts. And since all of us, the authors, participated in organizing the photos and designing the survey, we were not suitable to participate in the survey ourselves as “experts”.

15. Line 242-246: this study potentially (please see my comments about testing uniqueness and consistency of body patterns above) showed that stripe patterns are stable over time in captivity in a fully standardized rearing environment since before birth. Could variable field conditions and injuries affect body patterns? This and similar potential limits of these results should be discussed.

Reply:

We recognize the limitation described by the reviewer and have since included the following detail:

Lines 189-192: “Per our observations, environmental disturbances can stimulate a change in the animal’s color morph, from dark to light, light to dark [Fig 2a], or flashing between morphs [Fig 2b]. Over the course of our study, we attempted to avoid disturbing and distressing the animals during feeding, tank transfer, photography sessions, etc.. ”

Lines 293-297: “It is conceivable that environmental disturbances and injuries may affect the stripe configuration of O. chierchiae in the wild. However, since so little is known about the species in its natural habitat and we avoided injuring animals as much as possible over the course of this study, we do not have conclusive evidence whether or not environmental conditions and injuries would affect an animal’s stripe configuration.” 

16. Lines 220-226: the “specific examination of the problematic questions” should be described in more details. How were the factors underlying the problematic questions identified? Was their influence explicitly tested or just hypothesised?

Reply:

The factors underlying the problematic questions were hypothesized based on common attributes and characteristics within their corresponding photographs. The influence of these hypothetical factors underlying the problematic questions were not explicitly tested. 

These hypothesized problematic questions were completely removed in our newly designed survey.

17. Ethics statement: several key information such as the field permit number (or if this is not needed or it is included in the formal waiver already), formal waiver number and details on animal welfare are missing. Research projects involving human participants includes the use of data collection methods which are not face-to-face such as online surveys unless they are analysed anonymously, but it is not clear if this was case. The appropriate details should be added here and in Materials and Methods as indicated by the journal.

Reply:

We have since added the following statement explicitly:

Lines 143-145: “An official statement from the UC Berkeley OLAC is provided in the supplementary materials [S1 File]. Human participants were recruited via social media and all data was analyzed anonymously.” 

Lines 235-239: “However, because our data collection methods were not face-to-face and were analyzed anonymously…”

Minor comments

18. Line 31: here and in the rest of the text, I would be more specific in indicating that body patterns refer to stripes’ configurations, as this study did not consider other features such as skin texture.

Reply:

We have implemented the change in terminology per reviewer recommendation. We have replaced all relevant mentions of “body pattern” with “stripe configuration”.

19. Line 34: the untrained volunteers survey was conducted on laboratory-reared animals, not wild ones. Please rephrase this sentence and the rest of the main text accordingly. Please also consider replacing “demonstrate” with “might suggest”.

Reply:

We have implemented the change in terminology.

Lines 35-37: “These results demonstrate that laboratory-reared individuals could be identified via photographs taken at different points in their lifetime, which might suggest wild individuals can also be recognized and observed for field longitudinal studies.”

20. Line 87-88: I would suggest replacing “innocuous” with “non-invasive”. I agree that data collection through photoidentification is less invasive than handling, but it likely still involves some stress, being observed/photographed by someone can be an unpleasant experience for humans too. I would avoid using strong words such as “harmless”, “non-intrusive” and instead indicate that such techniques are less impactful on animal welfare.

Reply:

We have implemented the change in terminology per reviewer comment.

21. Line 35, 92, 251, 263: “in situ“ should be in italics. Alternating or replacing it with “field” might be clearer to readers.

Reply:

We have implemented the change in terminology per reviewer suggestion. We have replaced some of the mentions of “in situ” in the manuscript with “field”.

22. Line 102: please add more recent references for photo identification methods.

Reply:

We have since included more recent citations within the past five years.

(Johnston et al., 2017; Schneider, 2019; Vidal, 2021)

Johnston DR, Rayment W, Slooten E, Dawson SM. A time-based method for defining associations using photo-identification. Behaviour. 2017;154(9–10):1029–50.

Schneider S, Taylor GW, Linquist S, Kremer SC. Past, present and future approaches using computer vision for animal re‐identification from camera trap data. Methods Ecol Evol. 2019;10(4):461–70. Available from: http://dx.doi.org/10.1111/2041-210x.13133

Tan HY, Goh ZY, Loh K-H, Then AY-H, Omar H, Chang S-W. Cephalopod species identification using integrated analysis of machine learning and deep learning approaches. PeerJ. 2021;9(e11825):e11825. Available from: http://dx.doi.org/10.7717/peerj.11825

Vidal M, Wolf N, Rosenberg B, Harris BP, Mathis A. Perspectives on individual animal identification from biology and computer vision. Integr Comp Biol. 2021;61(3):900–16. Available from: http://dx.doi.org/10.1093/icb/icab107

23. Line 134: what is the age of sexual maturity?

Reply:

We have since included the following sentence:

Lines 156-158: “As gonad maturity is not currently defined in cephalopods, we approximate the age indicative of sexual maturity to be 16 weeks old, based on behavioral observations, i.e. engaging in copulation.”

24. Lines 140-141: please explain the relevance of analysing a generation bred and raised in captivity.

Reply:

We have since included the following sentence explaining the relevance of analyzing a generation of animals bred and raised in captivity:

Lines 167-170: “While the exact age and influence of life experiences on the development of wild caught individuals cannot be determined, having study animals bred and raised entirely in captivity allowed us the unique opportunity to follow the species’ development throughout its entire lifecycle, hatching till death.”

25. Lines 141-146: do these numbers refer to the F2 alone or include the F1?

Reply:

This number refers to both F1 and F2 combined. 

We have since removed all references to generations (F0, F1, F2) from the main body of text, as per reviewer comments, and moved these more detailed breeding and rearing information to the Supplementary Materials [S7 Supporting Information].

26. Lines 136-137: are all mates from the F1 generation?

Reply:

No, not all mated pairs were from the F1 generation. Some breeding pairs were chosen including wild caught individuals in order to reduce inbreeding. We have removed all reference to generations (F0, F1, F2) from the main body of text, as per reviewer comments, and moved these more detailed breeding and rearing information to the Supplementary Materials [S7 Supporting Information].

27. Lines 179-180: please specify if “this photobank” refers to pictures collected in this study or in the mentioned reference.

Reply:

Photobank refers to the pictures collected in this study

Lines 218-219: “We created a survey following the methods of Huffard et. al. (2008) using the photobank of images taken over the course of our study.”

28. Line 185: the pattern tracers were superimposed, not below the photo. Please be aware that some visually impaired (e.g., colour-blind) people might have difficulties in seeing them.

Reply: 

The pattern traces referenced in this reviewer comment were not superimposed. 

We understand and acknowledge that some visually impaired people may have difficulties in seeing the pattern traces. In the future, a research effort with more dedicated time and resources would ideally use color-blind friendly palettes to create the pattern traces, thus making them more accommodating for those who are visually impaired.

29. Lines 191-198: the main text, specifically the Methods, should include this important information that is currently disclosed only these captions, i. e., the pictures pairs showed either the same individual on different days or different individuals on the same day.

Reply:

Due to us having replaced our previous surveys with a newly designed survey, all the captions referenced in this comment and their associated figures have since been removed from the manuscript.

30. Lines 202-204: please specify if experts were able to recognize individuals thanks to the uniqueness or/and consistency of body patterns.

Reply:

At the time of this study, there were no established experts on this species aside from our research group, that we know of. We have since included the following clarification in our manuscript:

Lines 242-244: “All members of our research group (the authors of this paper) were able to recognize individual animals based on a combination of the uniqueness and consistency of the octopuses’ stripe configurations.”

31. Line 240: the survey C was not mentioned before.

Reply:

Survey C has since been removed from our study.

32. 267-268: please report all the information in Methods, including the age at which photo recording started. Why were animals not photographed until they were at least four weeks old?

Reply:

We have since included this information to the Methods section:

Lines 186-189: “The animals were not photographed until they were at least four weeks old because that is the earliest age when every individual’s stripe configuration became clearly discernible on the phone cameras we used to take the photos.”

33. Discussion: I would be more cautious in using expressions such as “our surveys confirm” (line 244), “suggest” would be more appropriate.

Reply:

We have implemented the recommended change in terminology by replacing “confirm”with “suggest” in the referenced line.

34. Line 255: “en masse” should be in italics.

Reply:

We have implemented the recommended correction.

35. Lines 250-287: I enjoyed reading these potential applications, which included a width of perspectives that should be more common in methodological studies.

Reply:

 We sincerely appreciate the reviewer’s kind words and comments.

36. Figure 1 and 2 caption: the species’ name should not start with a capital letter and the complete name should be in italics, as reported elsewhere in the manuscript.

Reply:

We have implemented the correction per reviewer comment.

37. Figure 1-2 and 6-9: I would suggest grouping these pictures in fewer figures (e.g., Figure 1 and 2 could become a single one with multiple panels).

Reply:

We have implemented the suggested changes in figure formats.

We have combined Figure 1 and 2 into Figure 1a and 1b.

Figure 6-9 of our original manuscript submission have been removed entirely. Our redesigned survey may be found in our Supplementary Materials [S4 File].

38. Figure 10 and 11: these figures and captions mentioned “scores” that were not explained in the main text. Please describe how these “scores” were obtained from the surveys’ responses in Methods.

Reply:

We have since included the following clarification:

Lines 231: “ Each set of survey responses was given a score [(correct answers)/(total questions) × 100]”

39. The number of figures is quite high and might overwhelm readers. I would suggest focusing on few, most relevant ones and moving the others to Supplementary.

Reply:

We have implemented the recommended change by the reviewer and reduced the number of figures by focusing only on a few relevant figures in the main body of text and moving the rest to Supplementary Materials or removing them entirely.

40. Data availability: please include a statement in the main text too and specify what type of data is available.

Reply:

We have since included the following statement detailing the type of data that is available:

252-253: “Survey data can be found in the Supplementary Materials [S5 Dataset].”

41. Numbers from zero to ten (in some cases to thirteen) are conventionally written using words and not numbers.

Reply:

We have implemented the change according to the reviewer comment.

Reviewer #2: 

1. The idea behind this paper is worthwhile yet the scholarship overall is subpar and there is one major problem. Both can be remedied if the authors concur - and want their research to be used and further developed by others. I provide examples below and give suggestions on how to address these. For example, the authors provide a very slanted choice of references to “support” their paper (sample details below).

The major flaw in the paper is that the authors do not understand that leucophores in the dermis are the basis of these permanent and species-specific white markings; in this case of O. chierchiae they are also the basis of individual specificity. Leucophores are not even mentioned in this paper.

Here is the basic scenario: chromatophores are mostly evenly distributed in the skin and when selectively activated create pigmentary patterns. Iridophores and leucophores (i.e. “white cells”) are not evenly distributed; they occur only in specific parts of the skin. Leucophores are aggregates of static cells (they do not have nerves, muscles etc) and are common in nearly all octopuses and cuttlefish. Andrew Packard and his colleagues as well as Roger Hanlon and colleagues have published widely on this and you need to reference some of those papers so that readers understand where this “fingerprinting” comes from. (For example: Packard, A., & Hochberg, F. G. (1977). Skin patterning in Octopus and other genera. Symposia of the Zoological Society of London, 38, 191-231. )

Both of those authors have shown that innervation patterns of chromatophores are aligned to expand adjacent to leucophore white markings to produce the high-contrast zebra-like markings. This was demonstrated neurophysiologcally and anatomically, for example, in Sepia officinalis (see Figs in Hanlon and Messenger 1988, Phil Trans). Conversely, other neural command neurons activate chromatophores directly over the leucophores to diminish (or mask) their bright whiteness.

By the way, leucophores produce some of the whitest white known in the animal kingdom (Mäthger, L. M., Senft, S. L., Gao, M., Karaveli, S., Bell, G. R. R., Zia, R., . . . Hanlon, R. T. (2013). Bright white scattering from protein spheres in color changing, flexible cuttlefish skin. Advanced Functional Materials, 23(32), 3980-3989 ) so this precise pigmentary masking by chromatophores is important and O chierchiae do this as demonstrated by your images.

Your Supplementary Fig. 10 in file S2_file.pdf shows this very well !! This figure has to be elevated to the main manuscript to illustrate what I explained above.

Reply:

We thank the reviewer for the information they have provided. We respectfully believe the topic of skin ultrastructure is beyond the scope of this study, and not necessary for the interpretation of our results. All of the skin components involved in octopus body color patterns are anatomically fixed in their location. While chromatophores are certainly involved, we do not feel it is necessary or appropriate to speculate about the degree to which iridophores and leucophores may or may not also be involved. 

We have included a summary of the recommended information and citations:

Lines 120-124: “Cephalopod body patterns are produced by a combination of pigment-filled chromatophores, color-reflective iridophores, and passively reflective leucophores, as well as muscular and hydrostatic forces that produce textural details (How et al., 2017; Mäthger et al., 2009) that operate within the fixed anatomical architecture of an octopus’ skin (Packard and Hochlberg, 1977; Messenger, 2001) [Fig 2].”

We have also moved Supplementary Fig. 10 from our original S2_file.pdf into the main body of text in our revised manuscript and renamed it Fig. 2b.

Specific Comments:

INTRODUCTION

2. Line 41 - delete this sentence - it adds nothing to the paper and such a statement applies to thousands of papers. Furthermore, the reference for this is inappropriate as it is written by someone who only studies octopuses.

Reply:

We have implemented this change and deleted the sentence.

3. L 47 the Hackett reference only covers the pop culture side of what is stated in this sentence. For the biological attributes supply a global rigorous study review such as Hanlon and Messenger 2018 book Cephalopod Behaviour or similar.

Reply:

We have since included a more recent scientific review article covering the same topic as the recommended book (discussing scientific interest in octopuses).

Di Cosmo A, Pinelli C, Scandurra A, Aria M, D’Aniello B. Research trends in octopus biological studies. Animals (Basel). 2021;11(6):1808. Available from: http://dx.doi.org/10.3390/ani11061808

4. L49. There are numerous studies have tracked individual cephalopods in the wild over time. It does not matter that it is difficult. You should cite 2-4 such studies - that have been done on octopuses, cuttlefishes, squids and nautilus!

(One of many examples: Hanlon, R. T., Forsythe, J. W., & Joneschild, D. E. (1999). Crypsis, conspicuousness, mimicry and polyphenism as antipredator defences of foraging octopuses on Indo-Pacific coral reefs, with a method of quantifying crypsis from video tapes. Biological Journal of the Linnean Society, 66(1), 1-22. Retrieved from ://000078601200001

Reply:

We have since included additional citations of studies tracking individual cephalopods in the wild overtime, more recent than the one the reviewer recommended.

Brewer RS, Norcross BL, Chenoweth E. Temperature- and size-dependent growth and movement of the North Pacific giant octopus (Enteroctopus dofleini) in the Bering Sea. Mar Biol Res. 2017;13(8):909–18.

Byrne RA, Wood JB, Anderson RC, Griebel U, Mather JA. Non-invasive methods of identifying and tracking wild squid. Ferrantia. 2010;59:22–31.

5. Overall this intro paragraph is heavily slanted and even a bit misleading. Kindly pose the study in a more appropriate and professional manner.

Reply: 

We appreciate the reviewer’s comment and have since removed several components of the introduction paragraph that may be perceived as misleading. We have also since included numerous citations, as per reviewer comments, to further establish the validity and professionalism of our writing. 

6. L 57. An appropriate reference for cephalopod semelparity is Rocha, F., Guerra, A., & Gonzalez, A. F. (2001). A review of reproductive strategies in cephalopods. Biological Reviews, 76(3), 291-304. doi:10.1017/s1464793101005681

Reply:

We have since added the recommended citation in Line 57.

7. L59 This statement of “… apart from most other octopus species” is nonsense. A very large number of octopus species produce very large eggs that hatch out as miniature adults (i.e. no paralarval/planktonic stage). The authors seem unaware of cephalopod life histories.

Reply:

Because the majority of octopus species with large eggs that hatch out as miniature adults are deep sea forms, we have made the following specification:

Lines 60-62: “Another trait that sets O. chierchiae apart from the majority of other shallow water octopus species (Sweeney et al., 1992) is its lack of a planktonic stage in its development, suggesting a limited potential for dispersal.”

We have also included the following citation detailing diverse cephalopod life histories to support our claim.

Sweeney MJ, Roper CFE, Mangold KM, Clark MR, Boletzky SV. “Larval” and juvenile cephalopods: a manual for their identification. Smithson Contrib Zool. 1992;(513):1–182. Available from: http://dx.doi.org/10.5479/si.00810282.513

8. L66 The Grearson et al 2021 is not even published … it is BioRivX … Moreover, this species is anything but a model organism! There are only a handful of journal papers and the species is nearly impossible to obtain from the wild, and only recently have lab trials begun in earnest. This sort of hyperbole is unacceptable.

Reply:

Grearson et al. 2021 has since been published. 

9. L 102 There are literally dozens of papers in which cephalopods have been studied with photography; at least cite a few of them to justify this paper and this Introduction.

Reply: 

 We have since included the following citations of cephalopods being studied with photography.

 Drerup C, Jackson A, Rickard C, Skea M, Cooke GM. Field observations on the behavioural ecology of the stout bobtail squid Rossia macrosoma (Cephalopoda: Sepiolidae) from Scottish waters. Mar Biodivers. 2021;51(4). Available from: http://dx.doi.org/10.1007/s12526-021-01202-y

 Vecchione M, Roper CFE. Cephalopods observed from submersibles in the western North Atlantic. Bull Mar Sci. 1991;49(13):433-45.

 Roper CFE, Hochberg FG. Behaviors and systematics of cephalopods from Lizard Island, Australia, based on color and body patterns. Malacologia. 1988;29(1):153-93

Tan HY, Goh ZY, Loh K-H, Then AY-H, Omar H, Chang S-W. Cephalopod species identification using integrated analysis of machine learning and deep learning approaches. PeerJ. 2021;9(e11825):e11825. Available from: http://dx.doi.org/10.7717/peerj.11825

10. L 104 Another method to follow individual octopus, squid and cuttlefish is scar tissue in the skin, or missing arm(s), or even marking the skin with dye in specific spots. These need to be mentioned for proper context in this paper.

Reply:

 We have since included details on these alternative methods in the following lines:

Lines 84-88: “Tattooing and branding techniques have had some success in cephalopod tracking studies as they eliminate the issue of tag removal and allow for small animals to be tagged (Semmens et al. 2007). However, they are often practically difficult to execute and may cause the animal severe distress and damage (Semmens et al. 2007).” 

Lines 114-117: “Longitudinal field studies conducted on the Caribbean reef squid, Sepiotheuthis sepioidea, have demonstrated that individual identification in the wild via body patterning and markings from injury (i.e. skin scar tissue, missing arms, miss-built arms), can be viable in cephalopods (Byrne et al., 2010).”

METHODS

11. This section if very long and includes many items that have nothing to do with this study. If you want credit for reading this species then do a separate ms and put in an appropriate mariculture journal. Otherwise shorten this section substantially. There are way too many figures in the main ms; if you choose to include so much culture info (which is not the biological basis of this paper) then at least put them in Supplementary sections.

Reply:

We have made the recommended reductions and moved a substantial amount of animal husbandry information and figures to the Supplementary Materials section [S2 Supporting Information]. We have also removed our survey figures and added references to a Supplementary file [S4 File] containing our entire survey.

RESULTS

12. L 201. How many individual octopuses were studied? The only hint of this is in the captions of Figs 7-9; from these the estimate would be perhaps as few as 2 octopuses. This is of central importance and the authors have to be crystal clear on this.

Reply:

The sample size of 25 individual octopuses is already stated in Lines 30-31: “photographically documenting the physical development of 25 octopuses from hatching” and Line 181 “[n = 25]”. 

We have reiterated this once more in the results section of our revised manuscript for greater clarity:

 Line 240: “[n = 25]” 

13. L 203 Where are the data on the results of the research group (this number has to be clear - do you mean the 5 coauthors?)

Reply:

To clarify our intent, the statement in our manuscript referenced by this reviewer comment is purely observational. We have since clarified that we mean the 5 coauthors. Considering that all 5 members contributed to the breeding and rearing of the animals as well as to photography and survey design, we did believe we could survey ourselves without introducing bias.

Line 242: “All members of our research group (the authors of this paper)…” 

14. Fig. 10 text/number letters are way too small. The y axis title of Percent of Total Score is ambiguous - can you provide a more intuitive title so that the graphic can stand alone in terms of understanding? … to that point, a color legend could/should be put directly on the graph … there is ample space amidst the blue bars.

Reply:

 We have made the suggested formatting changes. 

15. An important omission: you need to verify that the white stripes are leucophores and that their distribution is distinctive. The simplist fast method is to do some Light microscopy of gross anatomy of the leucophore stripes. This will give authenticity to your study. You can also illustrate this by taking a skin sample and photograph it normally then with light behind it to show that the leucophores stand out and block light in specific patterns. See Figs 14,15 i (Hanlon, R. T., & Messenger, J. B. (1988). Adaptive coloration in young cuttlefish (Sepia officinalis L.): The morphology and development of body patterns and their relation to behaviour. Philosophical Transactions of the Royal Society of London B, 320, 437-487. Note also Figs 18,19 the same paper to show how chromatophore expansion either accentuates or masks the whiteness.

Reply: 

We appreciate the reviewer’s comment and have since made the additions below to our manuscript. Please also see our reply to the other leucophore comment above.

Line 120-124: “Cephalopod body patterns are produced by a combination of pigment-filled chromatophores, color-reflective iridophores, and passively reflective leucophores, as well as muscular and hydrostatic forces that produce textural details (How et al., 2017; Mäthger et al., 2009) that operate within the fixed anatomical architecture of an octopus’ skin (Packard and Hochlberg, 1977; Messenger, 2001) [Fig 2].”

A microscopic study of the cellular makeup of the animal’s skin may be possible for other research groups, but adding this component would be well beyond the scope of this study, and would require sacrificing our study subjects. It is also not necessary to understand skin ultrastructure in order to understand the photoidentification process and give authenticity to this study (see Huffard et al. 2008). We have included the following line encouraging future interested researchers to pursue this study subject in the future:

Lines 325-329: “Identifying the specific anatomical composition of O. chierchiae’s stripe configuration was beyond the scope of our study, and would have required sacrificing our study subjects. We encourage future research efforts to be directed into examining the skin histology and ultrastructure of O. chierchiae, when deceased individuals are available, in order to gain further insight into the physiological basis of their stripe configuration development.”

DISCUSSION

16. L 217. These results are not very convincing - 75% correct is only fair to begin with, but even then only a bit more than half even scored that high.

Reply:

As per reviewer #1’s comment, we have modified our survey, correcting for problems that arose during our preliminary survey and attained new, more convincing results.

Lines 247-249: “Survey respondents (n = 38 responses) had a mean score = 84.2% with standard deviation = 15.36% and standard error of the mean = 2.49%. Survey respondents had a median score of 90%, with 44.74% of respondents (n = 17) scoring 95% or higher. [Fig 6].”

17. Moreover, the whole treatise seems to be based on 2 (or a very few) octopus individuals. (See comments in Results; perhaps there is a better explanation about how many octopuses were analyzed in this study).

Reply:

25 individual octopuses were used in this study, as stated in Lines 30-31: “photographically documenting the physical development of 25 octopuses from hatching” and Line 181 “[n = 25]”. 

We have reiterated this once more in the results section of our revised manuscript for greater clarity for future readers

Line 240: “[n=25]” 

18. L 225 “when a clear view of the body patterns is provided.” This is a large caveat to the study - your images in the lab were not all that useful/clear either, so going on later to say this could be useful in the field is a quantum leap in difficulty and utility.

Reply: 

We sincerely appreciate the reviewer for pointing out that this original sentence did not convey what we were intending to express accurately. We have since included the following clarification:

Lines 275-276: “when an unobstructed, roughly in-focus view of the stripe configuration on the mantle’s dorsal surface is provided.”

This caveat is in line with the typical quality control any researcher would conduct on their raw image data.

19. Overall, it appears that using such body patterns is quite difficult and requires considerable study and diligence by the biologists. The authors never really say this in this paper. Need to be more straight up on what the results actually indicate.

Reply: 

We respect and appreciate the concern raised by the reviewer. Based on the other inputs that we have received from the reviewers, we have since redesigned our survey. And the results of this redesigned survey are indicative that using stripe configurations are fairly easy and straightforward. Untrained volunteers attained high scores, with over 71% of survey respondents scoring 80% or higher on the survey, and a median score of 90%.

20. L 250+ Applying this technique to following individuals underwater will be very difficult (this reviewer has studied many octopus species in the wild; with many volunteers). The photography techniques in situ will require great diligence and patience to even get sufficient images to verify identity, much less acquire quantitative behavioral data.

Reply: 

We recognize and appreciate the concern raised by the reviewer. While we acknowledge that the application of this technique may be difficult for some researchers, one of our co-authors (Dr. Roy Caldwell) and one of our resources (Dr. Christine Huffard) have extensive experience successfully photographing and tracking wild individual octopuses underwater. They confirm that many researchers, and far more underwater photographers (both professional and amature) have sufficient skills to take these photographs. One need only conduct a google image search for “Blue Ringed octopus” or the smaller, far more cryptic “Hairy octopus” to appreciate the incredible talent in the underwater photography community. Thus we affirm that it is reasonable to believe such methods would be a viable pursuit for sufficiently skilled researchers, especially who are willing to form collaborations with underwater photographers (e.g. Huffard et al. 2008) if access to O. chierchiae’s habitat is available. 

21. L 244 To say you have no evidence of pattern change over time is a strong statement given how few animals were used.

Reply:

Based on reviewer #2’s comment 12 and 17, this reviewer seems to be under the impression that our study only consisted of 2 individuals. We have clarified that 25 animals were tracked over the 2 year course of this study..

22. L 245-249 Workers with some octopus and cuttlefish species have often used White frontal spots in octopus and details of zebra stripes in Sepia officinalis as species markers, and I believe J Boal made a comment on individual identity in S. officinalis in one of her pubs (you can search that one). Better off to give broad story and include leucophore markings in other cephs not just your octopus species.

Reply:

We have added the recommended citation:

Lines 288-291: “Although individual recognition has been explored in other cephalopods (Boal, 2006; Shashar 2004; Tricarico et al., 2011), to our knowledge, this is the first evidence that individually unique stripe configurations can be used by volunteers to consistently identify the same individual octopus throughout its development, from juvenile to adult.”

Boal J. Social recognition: a top down view of cephalopod behaviour. Vie et Milieu - Life & Environment. 2006;56(2):69-79.

Shashar N, Vaughan K, Loew E, Boal J, Hanlon R, Grable M. Behavioral evidence for intraspecific signaling with achromatic and polarized light by cuttlefish (Mollusca: Cephalopoda). Behaviour. 2004;141(7):837–61. Available from: http://dx.doi.org/10.1163/1568539042265662

Tricarico E, Borrelli L, Gherardi F, Fiorito G. I know my neighbour: individual recognition in Octopus vulgaris. PLoS One. 2011;6(4):e18710.

23. This paper has potential but requires major revision. These comments are given in a constructive manner to improve the science.

Reply

We have taken these comments into consideration during our revisions.

---

## [Decision Letter · Decision Letter 1]

17 Jan 2023

PONE-D-22-05774R1Individually unique, fixed stripe configurations of *Octopus chierchiae* allow for photoidentification in long-term studiesPLOS ONE

Dear Dr. Song,

Thank you for submitting your revised manuscript to PLOS ONE. After careful consideration, we feel that it has merit but does not fully meet PLOS ONE’s publication criteria as it currently stands. Therefore, we invite you to submit a revised version of the manuscript that addresses the points raised during the review process and below.

1) Please read over the comments of the reviewer, and address their concerns or explain why you don't.2) The reviewer asks you to identify patterns as per the categories of Packard & Hochberg. Is there some unique character as described in that paper that people are using to discern individuals? 3) In the first submission, you had two trials. In this paper, you have a third trial, but you have thrown out all your previous data. I don't understand how you can do that. Was that third trial run this past year after you received the reviews for this manuscript? Were any of the individuals 'taking the test' the same? That would mean they had more practice. Minimally, that information needs to be included in the appendix.4) Please be more specific about solicitation via social media.5) The reviewer asks for "corroborating the results with other evidence". One way to compare your results would be to have trained observers take the same test. You state "All members of our research group (the authors of this paper) were able to recognize individual animals based on a combination of the uniqueness and consistency of the octopuses stripe configurations", but did you ever test that?6) In Huffard et al. (2008), the conclusion was that people needed training in order to accurately identify individuals. Your results corroborate that conclusion, yet you state "to our knowledge, this is the first evidence that individually unique stripe configurations can be used by volunteers to consistently identify the same individual octopus throughout its development, from juvenile to adult." I would disagree with your statement. Rather, training is needed to reach an acceptable level (99.9%) of identification. I'm not sure why you have overstated your results, but that could be one reason the reviewers have asked for so much corroborating evidence to support your conclusions. 7) Overall the manuscript is too wordy. It can be made more succinct.8) See the instructions for PLOS ONE figure legend style. https://journals.plos.org/plosone/s/figures#loc-captions9) Please make sure that figures are numbered in the order they appear in the text. For example, your current order for figure three is incorrect (C comes before A and B).10) Give volumes for the chambers in figure three legend.11) Please stick with third person in the acknowledgements.12) In the literature cited section, please put all the genus/species names in italics.

We look forward to receiving your revised manuscript.

Kind regards,

Erik V. Thuesen, Ph.D.

Academic Editor

PLOS ONE

Reviewers' comments:

Reviewer's Responses to Questions

**Comments to the Author**

1. If the authors have adequately addressed your comments raised in a previous round of review and you feel that this manuscript is now acceptable for publication, you may indicate that here to bypass the “Comments to the Author” section, enter your conflict of interest statement in the “Confidential to Editor” section, and submit your "Accept" recommendation.

Reviewer #1: (No Response)

2. Is the manuscript technically sound, and do the data support the conclusions?

Reviewer #1: Partly

3. Has the statistical analysis been performed appropriately and rigorously? 

Reviewer #1: N/A

4. Have the authors made all data underlying the findings in their manuscript fully available?

Reviewer #1: Yes

5. Is the manuscript presented in an intelligible fashion and written in standard English?

Reviewer #1: Yes

6. Review Comments to the Author

Reviewer #1: I recognize the efforts of Authors in addressing Reviewers comments and suggestions. However, I found the revised version of the manuscript still with some pitfalls that does not put me in the condition to recommend for publication at this stage.I believe in the Authors view, that individuals of O. chierchiae exhibit unique patterns that allow identification of each individual animal.However, the approach and data provided do not support such an evidence with adequate robustness. The option of independent observers, survey is only one possibility and needs to be corroborated by other approaches.

As major issues:

1. I am unable to find a description of the different patterns exhibited by the individuals studied (and included in this ms), i.e. a description of patterns (chronic I assume, sensu Packard and Hochberg, 1977 - in the classic definition of body patterning of cephalopods, that the Authors seem to not consider at its fully extent). These descriptions need to be corroborated by some statistics (descriptive and correlation, at least). There is the need of bringing to the Readers objective evidence on the distinctiveness and stability of patterns observed.

2. I understand (from the comments to Reviewers) that the Authors prefer not to use computer assisted photo identification softwares at this stage, but there are other approaches that may allow to support their findings.

3. Thus, there is the need of corroborating the results with other evidence apart from the outcome from the survey circulated among untrained volunteers. This is only providing part of the evidence that Authors want to confirm their view.

4. Text in Lines 266-276 (and relevant figures) should be moved to Results5. Sentence at Lines 288-291 is misleading and scientifically inaccurate. Authors wrote "Although individual recognition has been explored in other cephalopods (Boal, 2006; Shashar 2004; Tricarico et al., 2011), to our knowledge, this is the first evidence that individually unique stripe configurations can be used by volunteers to consistently identify the same individual octopus throughout its development, from juvenile to adult". Individual recognition is a biological phenomenon, based on a cognitive feature of some animals, and used to describe the evidence that a given individual animal is capable of identifying a conspecific. This is not the case of this study. Authors, a different species, are using untrained volunteers (and the same is claimed in the text and in the comments to Reviewers, to be the case of caretaker and people involved in this study) - again a different species -, to identify individuals belonging to the target species. What Authors are adopting is a biometric identification of individual octopuses, based on features of chronic (or stable) patterns exhibited by the same animal over time.The sentence should be rephrased or deleted.

In addition, in several instances the Authors do not rely on accurate use of in-text citation, or the use of not adequate references is preferred to better and more classic/authoritative citations. For example:

Line 46 - aspects of the biology of the species/cephalopods are available in e.g., FAO volume 3 for octopuses (Jereb, P., Roper, C., Norman, M., and Finn, J. (2016). Cephalopods of the World. An Annotated and Illustrated Catalogue of Species Known to Date. Volume 3. Octopods and Vampire Squids. Roma, Italy: FAO, Food and Agriculture Organization of the United Nations). The citations included are not adequate, to my view and expertise.Similarly at Line 48 - Authors have various useful, authoritative options better than the two utilized and this Reviewer may eventually provide better suggestion.In many other instances the citations provided are only a very limited example of the possibilities. Thus, Authors may use e.g., for each of the in-text citations (otherwise do not supporting adequately the sentences).Line 58 - citation of Mather (2006) is useless and does not add any other information to the knowledge provided by the very good review by Rocha et al. (2001) that Authors also cite. The sentence should be rewritten and Mather (2006) is suggested to remove it from this context.The paragraph (Lines 56-62) needs rewriting. Authors may consider adding other species as example (e.g., O. bimaculoides; O. maya) in the same condition of this one. The cited work (Sweeney et al., 1992) provide a tabularized list of species with hatchlings so-called miniature adults (linked to low number of eggs from mothers and large amount of yolk).Similarly, in text citations at Lines 80 and 100 require an e.g.

Line 71. Authors refer to what is termed (appropriate definition) daily monitoring following Directive 2010/63/EU; see also requirements from ARRIVE Guidelines (refer for example to:

Kilkenny, C., Browne, W.J., Cuthill, I.C., Emerson, M., and Altman, D.G. (2010). Improving Bioscience Research Reporting: The ARRIVE Guidelines for Reporting Animal Research. PLOS Biology 8(6), e1000412. doi: 10.1371/journal.pbio.1000412.

Percie du Sert, N., Ahluwalia, A., Alam, S., Avey, M.T., Baker, M., Browne, W.J., Clark, A., Cuthill, I.C., Dirnagl, U., Emerson, M., Garner, P., Holgate, S.T., Howells, D.W., Hurst, V., Karp, N.A., Lazic, S.E., Lidster, K., MacCallum, C.J., Macleod, M., Pearl, E.J., Petersen, O.H., Rawle, F., Reynolds, P., Rooney, K., Sena, E.S., Silberberg, S.D., Steckler, T., and Würbel, H. (2020). Reporting animal research: Explanation and elaboration for the ARRIVE guidelines 2.0. PLOS Biology 18(7), e3000411. doi: 10.1371/journal.pbio.3000411.Fiorito, G., Affuso, A., Anderson, D.B., Basil, J., Bonnaud, L., Botta, G., Cole, A., D'Angelo, L., de Girolamo, P., Dennison, N., Dickel, L., Di Cosmo, A., Di Cristo, C., Gestal, C., Fonseca, R., Grasso, F., Kristiansen, T., Kuba, M., Maffucci, F., Manciocco, A., Mark, F.K., Melillo, D., Osorio, D., Palumbo, A., Perkins, K., Ponte, G., Raspa, M., Shashar, N., Smith, J., Smith, D., Sykes, A., Villanueva, R., Tublitz, N., Zullo, L., and Andrews, P.L.R. (2014). Cephalopods in neuroscience: Regulations, Research and the 3Rs. Invert. Neurosci 14, 13-36.

Fiorito, G., Affuso, A., Basil, J., Cole, A., de Girolamo, P., D'Angelo, L., Dickel, L., Gestal, C., Grasso, F., Kuba, M., Mark, F., Melillo, D., Osorio, D., Perkins, K., Ponte, G., Shashar, N., Smith, D., Smith, J., and Andrews, P.L. (2015). Guidelines for the Care and Welfare of Cephalopods in Research -  A consensus based on an initiative by CephRes, FELASA and the Boyd Group. Lab. Anim. 49(2 Suppl), 1-90.The sentence should be rewritten

7. PLOS authors have the option to publish the peer review history of their article (what does this mean?). If published, this will include your full peer review and any attached files.

Reviewer #1: No

---

## [Author Response · Author response to Decision Letter 1]

3 Mar 2023

3 March 2023

Dear Dr. Erik V. Thuesen,

 We sincerely appreciate your time and attention to our manuscript, and the constructive comments from the reviewer. We are grateful for the opportunity to address the issues with our resubmission. In addition to our revised manuscript and figure files, we also submit the manuscript file with track changes incorporated, for your reference if needed (Revised Manuscript with Track Changes.docx). 

Editor’s Comments:

1) Please read over the comments of the reviewer, and address their concerns or explain why you don't.

Reply:

 We have read over the comments of the reviewer and have addressed their concerns or provided our explanations to the best of our ability.

2) The reviewer asks you to identify patterns as per the categories of Packard & Hochberg. Is there some unique character as described in that paper that people are using to discern individuals? 

Reply:

 The pattern categories outlined in Tables III and IV of Packard and Hochberg (1977) include: White Mantle Spots, White Head-Bar, Frontal White Spots, Arm White Spots, Dark Arm Bar, Eye-Bar, Extended Eye-Bar, Dark Eye-Ring, Eyes-Spot (ocellus), Four Long Mantle Papillae, and Other. People may use some or all of these pattern categories, or something else entirely, to discern individual O. chierchiae. 

 As stated in Packard and Sanders (1971): “...if we ask ‘How many patterns are there in the octopus?’ the best, though hardly satisfactory, answer is ‘There are as many patterns as can be recognized by the classifier’ [...] the primary observational datum is the pattern, not the parts of which it is made up of." (Packard and Sanders, 1971)

 We have included references to Packard and Hochberg (1977) as well as Packard and Sanders (1971), to provide readers with more classic terminology for observed pattern components. However, per Packard and Sanders (1971), we believe that attempts to verbalize what each person may use as unique characters to discern individuals would be a pointless endeavor.

 Lines 197-200: “...pattern is generally composed of brown and white head bars, eye bars, extended eye bars, and transverse (chevron) stripes on the head and mantle, and longitudinal stripes on the arm crown. Occasionally, small brown mantle dark spots can be found within a white mantle stripe, and vice versa. The precise shape of each of these components was variable between individuals ”

3) In the first submission, you had two trials. In this paper, you have a third trial, but you have thrown out all your previous data. I don't understand how you can do that. Was that third trial run this past year after you received the reviews for this manuscript? Were any of the individuals 'taking the test' the same? That would mean they had more practice. Minimally, that information needs to be included in the appendix.

Reply:

 The third trial run this past year was performed after we had received the reviews for our initial manuscript submission. This was done in consideration of the reviewer comments recommending us to refine our survey and see if the results improve. 

 We neglected to include this information on our previous submission and have now created Supplemental Material S5 and S6 where we provide the original surveys, results, description of the problems found, and how we improved upon them for the redesigned trial reported in the main body of the paper.

 We recognize the concern that if the same individuals were to take the survey again, they may have additional practice and thus bias the result. Consequently, in our social media solicitation, we publicized and advertised this new survey on social media platforms and channels not used in our previous trials specifically to reach out to people who had not completed the previous two surveys. We also noted in the solicitation message requesting people who had taken the previous two surveys not to participate in this third redesigned survey. As a result, we have reason to believe that none of the individuals taking the test were the same. 

4) Please be more specific about solicitation via social media.

Reply:

 We reached out to Slack channels and Facebook groups that do not contain individuals who had participated in the previous two surveys and requested volunteers to complete our third revised survey. These Slack channels and Facebook groups range from our workplace channels to hobbyist groups that are not biology or science oriented in order to minimize bias in participant backgrounds. Additionally, we specifically stated in our solicitation message asking people who had taken the previous two surveys not to participate in this third redesigned survey in an effort to eliminate bias from additional practice.

 We further specified our social media solicitation methods in Lines 141-144: “Human participants were recruited via social media by reaching out in Slack channels and Facebook groups. These social media channels ranged from our workplace channels to hobbyist groups that are not biology or science oriented in order to minimize bias from participant backgrounds.”

5) The reviewer asks for "corroborating the results with other evidence". One way to compare your results would be to have trained observers take the same test. You state "All members of our research group (the authors of this paper) were able to recognize individual animals based on a combination of the uniqueness and consistency of the octopuses stripe configurations", but did you ever test that?

Reply:

 All authors of this paper took the survey and were able to recognize individual animals with 100% accuracy. We did not explicitly test this as we were all involved in the creation of the survey itself, which makes taking the survey ourselves an uninformative and unrigorous endeavor.

 Additionally, as we mentioned in our previous review response, due to the extremely limited access to O. chierchiae, there are simply too few people in the world who are adequately trained and experienced with the species outside of our own research group to take the survey as “trained observers” and make a meaningful comparison with the results of untrained volunteers.

6) In Huffard et al. (2008), the conclusion was that people needed training in order to accurately identify individuals. Your results corroborate that conclusion, yet you state "to our knowledge, this is the first evidence that individually unique stripe configurations can be used by volunteers to consistently identify the same individual octopus throughout its development, from juvenile to adult." I would disagree with your statement. Rather, training is needed to reach an acceptable level (99.9%) of identification. I'm not sure why you have overstated your results, but that could be one reason the reviewers have asked for so much corroborating evidence to support your conclusions. 

Reply:

 In Huffard et al. (2008), the authors explicitly stated in their discussion: "Untrained observers were able to differentiate individual W. photogenicus based on photographed body patterns." The results of our study corroborate that conclusion.

 In addition, we do not know of any formal standard that defines the acceptable level of photoidentification as 99.9% accurate. In fact, nearly all of the published photoidentification studies involving untrained observers that we examined reported less than 99.9% accuracy, usually involving a smaller sample size of volunteers than ours as well (Huffard et al., 2008; Cooley et al., 2013; Chaves et al., 2016; Treilibs et al., 2016; Theimer et al., 2017). The expectation of 99.9% accuracy is unrealistic and not representative of working expectations in actual photoidentification studies. By contrast, marine animal photoidentification projects involving machine learning often provide a correct match less than 80% of the time, and yet can still be very useful (Katija et al., 2022).

Huffard CL, Caldwell RL, DeLoach N, Gentry DW, Humann P, MacDonald B, et al. Individually unique body color patterns in octopus (Wunderpus photogenicus) allow for photoidentification. PLoS One. 2008;3(11):e3732.

Cooley CR, Smith SA, Geier CJ, Puentes TG. The use of photo-identification as a means of identifying Western Painted Turtles (Chrysemys picta bellii) in long-term demographic studies. Herpetol. Rev. 2013;44(3):432–434.

Chaves LCT, Hall J, Feitosa JLL, Côté IM. Photo-identification as a simple tool for studying invasive lionfish Pterois volitans populations: Pterois volitans photo-identification. J Fish Biol [Internet]. 2016;88(2):800–4. Available from: http://dx.doi.org/10.1111/jfb.12857

Treilibs CE, Pavey CR, Hutchinson MN, Bull CM. Photographic identification of individuals of a free‐ranging, small terrestrial vertebrate. Ecol Evol [Internet]. 2016;6(3):800–9. Available from: http://dx.doi.org/10.1002/ece3.1883

Theimer TC, Ray DT, Bergman DL. Camera Angle and Photographic Identiﬁcation of Individual Striped Skunks. Wildl. Soc. Bull. 2017;41(1):146–150.

Katija K, Orenstein E, Schlining B, Lundsten L, Barnard K, Sainz G, et al. FathomNet: A global image database for enabling artificial intelligence in the ocean. Sci Rep [Internet]. 2022;12(1):15914. Available from: http://dx.doi.org/10.1038/s41598-022-19939-2

7) Overall the manuscript is too wordy. It can be made more succinct.

Reply:

 We recognize this concern and have since made the writing of our manuscript more succinct where possible.

8) See the instructions for PLOS ONE figure legend style. https://journals.plos.org/plosone/s/figures#loc-captions

Reply:

 We have since corrected the formatting of our figure legends according to the instructions provided.

9) Please make sure that figures are numbered in the order they appear in the text. For example, your current order for figure three is incorrect (C comes before A and B).

Reply:

 We have since corrected this mistake and all of the figures are now numbered in the order they appear in the text.

10) Give volumes for the chambers in figure three legend.

Reply:

 We have now provided the volumes for the chambers in Figure 3’s legend.

11) Please stick with third person in the acknowledgements.

Reply:

 We have since corrected this mistake and now use third person exclusively in the acknowledgments.

12) In the literature cited section, please put all the genus/species names in italics.

Reply:

 We have since corrected this mistake and have put all the genus/species names in the literature cited section in italics.

Reviewers' comments:

Reviewer's Responses to Questions

Comments to the Author

1. If the authors have adequately addressed your comments raised in a previous round of review and you feel that this manuscript is now acceptable for publication, you may indicate that here to bypass the “Comments to the Author” section, enter your conflict of interest statement in the “Confidential to Editor” section, and submit your "Accept" recommendation.

Reviewer #1: (No Response)

2. Is the manuscript technically sound, and do the data support the conclusions?

Reviewer #1: Partly

3. Has the statistical analysis been performed appropriately and rigorously?

Reviewer #1: N/A

4. Have the authors made all data underlying the findings in their manuscript fully available?

Reviewer #1: Yes

5. Is the manuscript presented in an intelligible fashion and written in standard English?

Reviewer #1: Yes

6. Review Comments to the Author

Reviewer #1: I recognize the efforts of Authors in addressing Reviewers comments and suggestions. However, I found the revised version of the manuscript still with some pitfalls that does not put me in the condition to recommend for publication at this stage. I believe in the Authors view, that individuals of O. chierchiae exhibit unique patterns that allow identification of each individual animal. However, the approach and data provided do not support such an evidence with adequate robustness. The option of independent observers, survey is only one possibility and needs to be corroborated by other approaches.

Reply:

 Based on several of the sources that we had examined (Bryne et al. 2010; Theimer et al., 2017), we believe that only pursuing one approach of surveying independent observers is sufficient for this initial study of the species. 

 The only other paper we know of on octopus photoidentification (Huffard et al. 2008, published in PLOSONE) was based on fewer individuals studied over a much shorter duration. We used similar photoidentification methods to those outlined in that paper, but involved a larger number of volunteers. Performance of our study subjects was comparable to that of Huffard et al. 2008.

 However, we appreciate the concern raised by the reviewer and have included Figure 4: a display of a single representative photograph of each individual included in our survey study; with accompanying S7 Dataset: a count of different pattern components captured by photograph and observed by our research team on the dorsal mantle surface of each individual. This figure and table provide additional evidence of the uniqueness of O. chierchiae patterns.

Byrne RA, Wood JB, Anderson RC, Griebel U, Mather JA. Non-invasive methods of identifying and tracking wild squid. Ferrantia. 2010;59:22–31.

Theimer TC, Ray DT, Bergman DL. Camera Angle and Photographic Identiﬁcation of Individual Striped Skunks. Wildl. Soc. Bull. 2017;41(1):146–150.

Huffard CL, Caldwell RL, DeLoach N, Gentry DW, Humann P, MacDonald B, et al. Individually unique body color patterns in octopus (Wunderpus photogenicus) allow for photoidentification. PLoS One. 2008;3(11):e3732.

As major issues:

1. I am unable to find a description of the different patterns exhibited by the individuals studied (and included in this ms), i.e. a description of patterns (chronic I assume, sensu Packard and Hochberg, 1977 - in the classic definition of body patterning of cephalopods, that the Authors seem to not consider at its fully extent). These descriptions need to be corroborated by some statistics (descriptive and correlation, at least). There is the need of bringing to the Readers objective evidence on the distinctiveness and stability of patterns observed.

Reply:

 We appreciate the reviewer’s comments and have since included descriptions of the different patterns exhibited by the individuals studied in the following lines:

 Lines 189-200: “Terms used to describe O. chierchiae body patterns follow those defined and illustrated previously (Packard and Sanders, 1969; Packard and Sanders, 1971; Packard and Hochberg, 1977, Caldwell et al., 2015). Morphological appearance of each animal may vary greatly depending on body position or chromatophore activation. Consequently, this investigation focused on the stripe configurations of the chronic patterns located on the animals’ dorsal mantle surface [Fig 4A], because these were the most consistently observable identifying characteristics. Chronic patterns are those that persist for hours or days, or those that the animal returns to after acute variations that last seconds or minutes (Packard and Sanders, 1969). The stripe-bar-spot pattern is generally composed of brown and white head bars, eye bars, extended eye bars, and transverse (chevron) stripes on the head and mantle, and longitudinal stripes on the arm crown. Occasionally, small brown mantle dark spots can be found within a white mantle stripe, and vice versa. The precise shape of each of these components was variable between individuals.”

 We have since incorporated Figure 4: a display of a single representative photograph of each individual included in our survey study; with accompanying S7 Dataset: a count of different pattern components captured by photograph and observed by our research team on the dorsal mantle surface of each individual. These further showcase the uniqueness of O. chierchiae body patterns. 

 We have also applied descriptive statistics as objective corroborating evidence using the S7 Dataset referencing Vaissi et al., (2018).

 Lines 200-207: “We chose to specifically examine [S7 Dataset] the dark transverse mantle stripes and dark mantle spots in the chronic patterns of O. chierchiae. We defined dark transverse mantle stripes as any dark pattern component that traverses the width of the individual’s mantle. We further categorized these into dark continuous stripes [Fig 4B] and dark discontinuous stripes [Fig 4C], those that contained no visible breakage and those that contained visible breakage, respectively. We also defined dark mantle spots [Fig 4D] as any dark pattern component completely encircled by white pattern components on an individual’s mantle.” 

 Lines 269-273: “The O. chierchiae bred and raised in our lab displayed [Fig 4] an average of 4.8 (±0.49 S.D.) dark transverse mantle stripes in their chronic patterns. In 20% (n =5) of the animals, this included one discontinuous stripe [Fig 4C]. Additionally, 24% (n =6) of the animals also displayed a dark mantle spot [Fig 4D] [S7 Dataset]. Examples of how pattern component counts were obtained may be found in Supplementary Materials [S8 Figure].”

 Correlation statistics are not applicable in our study. As we observed stable and consistent stripe configurations over time, there are not two variables to draw a correlation between. For example, if the number of stripe components remains consistent, that would be a constant. By the statistical definition of correlation, there must be two variables to consider. Attempts to apply correlation statistics such as Pearsons or Spearman's correlation resulted in undefined correlations. However, we believe that our timeline figures [Fig 5] adequately demonstrate O. chierchiae’s body pattern stability over extended time spans. 

Vaissi S, Farassat H, Akmali V, Sharifi M. Consistency of coloration pattern and applicability of photo identification method as a tool to identify individuals of the kaiser’s mountain newt, Neurergus kaiseri. Russ J Herpetol. 2018;25(4):311.

2. I understand (from the comments to Reviewers) that the Authors prefer not to use computer assisted photo identification softwares at this stage, but there are other approaches that may allow to support their findings.

Reply:

 We have since incorporated Figure 4: a display of a single representative photograph of each individual included in our survey study; with accompanying S7 Dataset: a count of different pattern components captured by photograph and observed by our research team on the dorsal mantle surface of each individual. These further showcase the uniqueness of O. chierchiae body patterns. 

 We have also applied descriptive statistics as objective corroborating evidence using the S7 Dataset referencing Vaissi et al., (2018).

 Lines 200-207: “We chose to specifically examine [S7 Dataset] the dark transverse mantle stripes and dark mantle spots in the chronic patterns of O. chierchiae. We defined dark transverse mantle stripes as any dark pattern component that traverses the width of the individual’s mantle. We further categorized these into dark continuous stripes [Fig 4B] and dark discontinuous stripes [Fig 4C], those that contained no visible breakage and those that contained visible breakage, respectively. We also defined dark mantle spots [Fig 4D] as any dark pattern component completely encircled by white pattern components on an individual’s mantle.” 

 Lines 269-273: “The O. chierchiae bred and raised in our lab displayed [Fig 4] an average of 4.8 (±0.49 S.D.) dark transverse mantle stripes in their chronic patterns. In 20% (n =5) of the animals, this included one discontinuous stripe [Fig 4C]. Additionally, 24% (n =6) of the animals also displayed a dark mantle spot [Fig 4D] [S7 Dataset]. Examples of how pattern component counts were obtained may be found in Supplementary Materials [S8 Figure].”

 We believe that our timeline figures [Fig 5] adequately demonstrate O. chierchiae’s body pattern stability over extended time spans. 

 We believe that the information presented in Figure 4, Figure 5 and the descriptive statistics, in conjunction with our survey results, provide the readers with adequate objective evidence of the distinctiveness and stability of O. chierchiae patterns which are identifiable by even untrained volunteers.

3. Thus, there is the need of corroborating the results with other evidence apart from the outcome from the survey circulated among untrained volunteers. This is only providing part of the evidence that Authors want to confirm their view.

Reply:

 We appreciate the reviewer’s comment. However, we believe that the survey results by untrained volunteers provide adequately robust support to our conclusion that O. chierchiae’s unique stripe configurations allow individual animals to be identified and distinguished by untrained observers as the same approach has been used to draw similar conclusions in Wunderpus photogenicus, Chrysemys picta bellii, Sepiotheuthis sepioidea, Mephitis mephitis, and etc. (Huffard et al., 2008; Bryne et al. 2010; Cooley et al., 2013; Theimer et al., 2017).

 We have since incorporated Figure 4: a display of a single representative photograph of each individual included in our survey study; with accompanying S7 Dataset: a count of different pattern components captured by photograph and observed by our research team on the dorsal mantle surface of each individual. These further showcase the uniqueness of O. chierchiae body patterns. 

 We have also applied descriptive statistics as objective corroborating evidence using the S7 Dataset referencing Vaissi et al., (2018).

 Lines 200-207: “We chose to specifically examine [S7 Dataset] the dark transverse mantle stripes and dark mantle spots in the chronic patterns of O. chierchiae. We defined dark transverse mantle stripes as any dark pattern component that traverses the width of the individual’s mantle. We further categorized these into dark continuous stripes [Fig 4B] and dark discontinuous stripes [Fig 4C], those that contained no visible breakage and those that contained visible breakage, respectively. We also defined dark mantle spots [Fig 4D] as any dark pattern component completely encircled by white pattern components on an individual’s mantle.” 

 Lines 269-273: “The O. chierchiae bred and raised in our lab displayed [Fig 4] an average of 4.8 (±0.49 S.D.) dark transverse mantle stripes in their chronic patterns. In 20% (n =5) of the animals, this included one discontinuous stripe [Fig 4C]. Additionally, 24% (n =6) of the animals also displayed a dark mantle spot [Fig 4D] [S7 Dataset]. Examples of how pattern component counts were obtained may be found in Supplementary Materials [S8 Figure].”

 We believe that our timeline figures [Fig 5] adequately demonstrate O. chierchiae’s body pattern stability over extended time spans. 

 We believe that the information presented in Figure 4, Figure 5 and the descriptive statistics, in conjunction with our survey results, provide the readers with adequate objective evidence of the distinctiveness and stability of O. chierchiae patterns which are identifiable by even untrained volunteers.

Huffard CL, Caldwell RL, DeLoach N, Gentry DW, Humann P, MacDonald B, et al. Individually unique body color patterns in octopus (Wunderpus photogenicus) allow for photoidentification. PLoS One. 2008;3(11):e3732.

Cooley CR, Smith SA, Geier CJ, Puentes TG. The use of photo-identification as a means of identifying Western Painted Turtles (Chrysemys picta bellii) in long-term demographic studies. Herpetol. Rev. 2013;44(3):432–434

Byrne RA, Wood JB, Anderson RC, Griebel U, Mather JA. Non-invasive methods of identifying and tracking wild squid. Ferrantia. 2010;59:22–31.

Theimer TC, Ray DT, Bergman DL. Camera Angle and Photographic Identiﬁcation of Individual Striped Skunks. Wildl. Soc. Bull. 2017;41(1):146–150.

4. Text in Lines 266-276 (and relevant figures) should be moved to Results

Reply:

 We have implemented the reviewer suggestion and moved the text in Lines 266-276 and any relevant figures to the Results section.

 Lines 290-297: “To investigate if there was a significant difference between false positive and false negative rates in the survey response, we performed the Wilcoxon Signed Rank Test with our null hypothesis being that false positive rates and false negative rates were the same, while our alternative hypothesis being that false positive rates and false negative rates were different. The test result revealed that there was not a significant difference between false positive and false negative rates (Wilcoxon Signed Rank Test: test statistic = 150, α = 0.05, n = 28, critical value = 116), suggesting a potential future study of wild populations would not be biased towards one error type or another.”

5. Sentence at Lines 288-291 is misleading and scientifically inaccurate. Authors wrote "Although individual recognition has been explored in other cephalopods (Boal, 2006; Shashar 2004; Tricarico et al., 2011), to our knowledge, this is the first evidence that individually unique stripe configurations can be used by volunteers to consistently identify the same individual octopus throughout its development, from juvenile to adult". Individual recognition is a biological phenomenon, based on a cognitive feature of some animals, and used to describe the evidence that a given individual animal is capable of identifying a conspecific. This is not the case of this study. Authors, a different species, are using untrained volunteers (and the same is claimed in the text and in the comments to Reviewers, to be the case of caretaker and people involved in this study) - again a different species -, to identify individuals belonging to the target species. What Authors are adopting is a biometric identification of individual octopuses, based on features of chronic (or stable) patterns exhibited by the same animal over time.The sentence should be rephrased or deleted.

Reply:

 We acknowledge that we had mistakenly misused the term “individual recognition” in Line 288-291 of our previous submission. We have since re-phrased this sentence and have moved the contained references into a more appropriate section of our manuscript:

 Lines 232-235: “To our knowledge, this is the first evidence that individually unique stripe configurations can be used by volunteers to consistently identify the same individual octopus throughout its development, from juvenile to adult”

 Lines 361-363: “Individual recognition has been explored in several other cephalopods (Shashar 2004; Boal, 2006), and at least one species, Octopus vulgaris, is thought to have the ability of individual conspecific recognition (Tricario et al., 2011). ”

In addition, in several instances the Authors do not rely on accurate use of in-text citation, or the use of not adequate references is preferred to better and more classic/authoritative citations. For example:

Line 46 - aspects of the biology of the species/cephalopods are available in e.g., FAO volume 3 for octopuses (Jereb, P., Roper, C., Norman, M., and Finn, J. (2016). Cephalopods of the World. An Annotated and Illustrated Catalogue of Species Known to Date. Volume 3. Octopods and Vampire Squids. Roma, Italy: FAO, Food and Agriculture Organization of the United Nations). The citations included are not adequate, to my view and expertise.

Reply:

 We appreciate the reviewer’s comment. However, in an effort to make our manuscript more concise, as per the editor’s suggestion, we have removed the text in question entirely.

Similarly at Line 48 - Authors have various useful, authoritative options better than the two utilized and this Reviewer may eventually provide better suggestion.In many other instances the citations provided are only a very limited example of the possibilities. Thus, Authors may use e.g., for each of the in-text citations (otherwise do not supporting adequately the sentences).

Reply:

 We appreciate the concern raised by the reviewer and have since utilized e.g. for many of our in-text citations. However, in an effort to make our manuscript more concise, as per the editor’s suggestion, we have removed some of the texts in question entirely.

 In the specific instance at Line 48, the citations included in this sentence were intended to refer to broad overviews of the wide array of possibilities which exists as part of our introductory hook. Therefore, we believe that the references provided here are adequate for our purposes.

Line 58 - citation of Mather (2006) is useless and does not add any other information to the knowledge provided by the very good review by Rocha et al. (2001) that Authors also cite. The sentence should be rewritten and Mather (2006) is suggested to remove it from this context.

Reply:

 We accept the reviewer’s suggestion and have since replaced the citation of Mather (2006) in Line 58 with a citation of the review by Rocha et al. (2001).

 Line 75-76: “While most octopus species exhibit semelparity and only reproduce near the end of their lifespan (Rocha et al., 2001)...”

The paragraph (Lines 56-62) needs rewriting. Authors may consider adding other species as example (e.g., O. bimaculoides; O. maya) in the same condition of this one. The cited work (Sweeney et al., 1992) provide a tabularized list of species with hatchlings so-called miniature adults (linked to low number of eggs from mothers and large amount of yolk).

Reply:

 We appreciate the reviewer’s comment. However, in an effort to make our manuscript more concise, as per the editor’s suggestion, we have eliminated the comparison of O. chierchiae’s reproductive strategy with those of other octopus species, as this is not the focus of our study.

Similarly, in text citations at Lines 80 and 100 require an e.g.

Reply:

 We have since rewritten these lines to include relevant examples for their respective in text citations:

 Lines 87-92: “Current studies on wild cephalopods most commonly involve tag-recapture studies (Semmens et al., 2007), as exemplified by efforts to investigate the growth and movement of Octopus vulgaris in Central Western Sardinia (Mereu et al., 2010) and Enteroctopus dofleini in the Bering Sea (Brewer et al., 2017). Visible Implant Elastomer tags have demonstrated viable usage in long term studies of larger cephalopods such as E. dofleini (Brewer and Norcross, 2012)”

 Lines 104-109: “The low-impact nature of this approach as well as the virtual ubiquitousness of high-quality cameras that can record photographs, videos, geospatial data, and temporal data, has enabled community science efforts to make increasingly significant contributions to scientific research, as showcased by the US National Parks Service’s What’s Invasive! project and NSF NEON’s Project BudBurst (Graham et al., 2011; Dickinson et al 2012; Chandler et. al., 2017).”

Line 71. Authors refer to what is termed (appropriate definition) daily monitoring following Directive 2010/63/EU; see also requirements from ARRIVE Guidelines (refer for example to:

Kilkenny, C., Browne, W.J., Cuthill, I.C., Emerson, M., and Altman, D.G. (2010). Improving Bioscience Research Reporting: The ARRIVE Guidelines for Reporting Animal Research. PLOS Biology 8(6), e1000412. doi: 10.1371/journal.pbio.1000412.

Percie du Sert, N., Ahluwalia, A., Alam, S., Avey, M.T., Baker, M., Browne, W.J., Clark, A., Cuthill, I.C., Dirnagl, U., Emerson, M., Garner, P., Holgate, S.T., Howells, D.W., Hurst, V., Karp, N.A., Lazic, S.E., Lidster, K., MacCallum, C.J., Macleod, M., Pearl, E.J., Petersen, O.H., Rawle, F., Reynolds, P., Rooney, K., Sena, E.S., Silberberg, S.D., Steckler, T., and Würbel, H. (2020). Reporting animal research: Explanation and elaboration for the ARRIVE guidelines 2.0. PLOS Biology 18(7), e3000411. doi: 10.1371/journal.pbio.3000411.Fiorito, G., Affuso, A., Anderson, D.B., Basil, J., Bonnaud, L., Botta, G., Cole, A., D'Angelo, L., de Girolamo, P., Dennison, N., Dickel, L., Di Cosmo, A., Di Cristo, C., Gestal, C., Fonseca, R., Grasso, F., Kristiansen, T., Kuba, M., Maffucci, F., Manciocco, A., Mark, F.K., Melillo, D., Osorio, D., Palumbo, A., Perkins, K., Ponte, G., Raspa, M., Shashar, N., Smith, J., Smith, D., Sykes, A., Villanueva, R., Tublitz, N., Zullo, L., and Andrews, P.L.R. (2014). Cephalopods in neuroscience: Regulations, Research and the 3Rs. Invert. Neurosci 14, 13-36.

Fiorito, G., Affuso, A., Basil, J., Cole, A., de Girolamo, P., D'Angelo, L., Dickel, L., Gestal, C., Grasso, F., Kuba, M., Mark, F., Melillo, D., Osorio, D., Perkins, K., Ponte, G., Shashar, N., Smith, D., Smith, J., and Andrews, P.L. (2015). Guidelines for the Care and Welfare of Cephalopods in Research - A consensus based on an initiative by CephRes, FELASA and the Boyd Group. Lab. Anim. 49(2 Suppl), 1-90.The sentence should be rewritten

Reply:

 We have since rewritten the sentence and have removed the term “monitor” from the text in question in our manuscript.

 Lines 83-86: “Specifically, the ability to identify and observe the octopuses on an individual basis would be crucial for both field studies (Arechavala-Lopez et al., 2019) as well as controlled laboratory research involving multiple individual animals (Morse et al., 2017).”

7. PLOS authors have the option to publish the peer review history of their article (what does this mean?). If published, this will include your full peer review and any attached files.

Do you want your identity to be public for this peer review? For information about this choice, including consent withdrawal, please see our Privacy Policy.

Reviewer #1: No

Once again, thank you for your time and attention as well as the constructive comments from the reviewer. In every case we have incorporated the recommended change or clarified our intentions and reasoning.

Sincerely,

Leo Song

University of California, Berkeley: Department of Integrative Biology 

3040 Valley Life Sciences Building # 3140

Berkeley, CA 94720-3200

leosong@berkeley.edu

(510) 570-5304

---

## [Editor Report · Decision Letter 2]

12 Mar 2023

Individually unique, fixed stripe configurations of *Octopus chierchiae* allow for photoidentification in long-term studies

PONE-D-22-05774R2

Dear Dr. Song,

We’re pleased to inform you that your manuscript has been judged scientifically suitable for publication and will be formally accepted for publication once it meets all outstanding technical requirements.

Kind regards,

Erik V. Thuesen, Ph.D.

Academic Editor

PLOS ONE
---

## [Editor Report · Acceptance letter]

22 Mar 2023

PONE-D-22-05774R2 

Individually unique, fixed stripe configurations of *Octopus chierchiae* allow for photoidentification in long-term studies 

Dear Dr. Song:

I'm pleased to inform you that your manuscript has been deemed suitable for publication in PLOS ONE. Congratulations! Your manuscript is now with our production department. 

Kind regards, 

on behalf of

Dr. Erik V. Thuesen 

Academic Editor

PLOS ONE